# A mechanistic multi-area recurrent network model of decision-making

**Michael Kleinman**[1]     **Chandramouli Chandrasekaran**[2]*     **Jonathan C. Kao**[1]*
[1]University of California, Los Angeles     [2]Boston University
michael.kleinman@ucla.edu    cchandr1@bu.edu    kao@seas.ucla.edu

## Abstract

Recurrent neural networks (RNNs) trained on neuroscience-based tasks have been widely used as models for cortical areas performing analogous tasks. However, very few tasks involve a single cortical area, and instead require the coordination of multiple brain areas. Despite the importance of multi-area computation, there is a limited understanding of the principles underlying such computation. We propose to use multi-area RNNs with neuroscience-inspired architecture constraints to derive key features of multi-area computation. In particular, we show that incorporating multiple areas and Dale's Law is critical for biasing the networks to learn biologically plausible solutions. Additionally, we leverage the full observability of the RNNs to show that output-relevant information is preferentially propagated between areas. These results suggest that cortex uses modular computation to generate minimal sufficient representations of task information. More broadly, our results suggest that constrained multi-area RNNs can produce experimentally testable hypotheses for computations that occur within and across multiple brain areas, enabling new insights into distributed computation in neural systems.

## 1    Introduction

Decision-making, multisensory integration, attention, motor control, and timing emerge from the coordination of multiple interconnected brain areas [1–9]. While neural activity in a particular area can contain behaviorally relevant signals, such as choices or percepts, it is often unclear if these signals originate within the area or are inherited from upstream brain areas [6]. Understanding the neural bases of these behaviors necessitates an understanding of the intra- and inter-area dynamics, that is, how neural activity evolves within and between brain areas. However, we currently lack clear hypotheses for how distinct brain-area dynamics and connectivity relate to computation. To address this gap, we use multi-area recurrent neural networks (RNNs) to model, probe, and gain insight into how behaviorally relevant computations emerge within and across multiple brain areas.

Optimized feedforward and recurrent neural networks have been used for machine learning but are also emerging tools to model computations associated with visual [10, 11], cognitive [12], timing [3, 13], navigation [14], and motor tasks [15, 16]. RNNs transform experimenter-designed task inputs into behavior-related outputs through recurrent dynamics. Its artificial units often exhibit heterogeneous responses and population dynamics observed in brain areas implicated in cognitive and motor tasks [12, 15, 17–20]. If artificial units resemble cortical neurons, RNNs are subsequently analyzed to propose hypotheses for how a local computation occurs in a brain area [3, 12, 18, 21]. In comparison to direct experimental recordings, an important advantage of RNNs is that the activity

---

*Joint senior author

35th Conference on Neural Information Processing Systems (NeurIPS 2021).

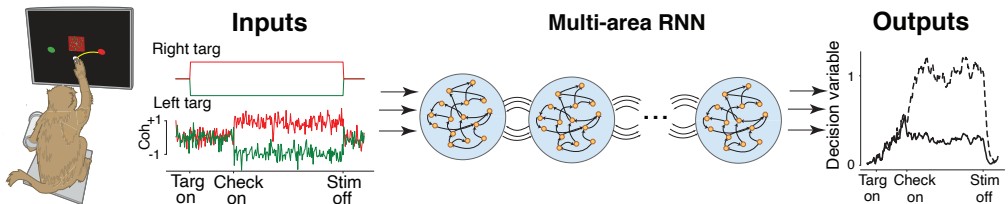

Figure 1: **Task.** RNN configuration. The RNN receives 4 inputs. Two inputs indicate the identity of the left and right targets, which can be red or green. These inputs are noiseless. The other two inputs indicate the value of the signed color coherence (proportional to amount of red in checkerboard) and negative signed color coherence (proportional to amount of green in checkerboard). We added independent Gaussian noise to these signals (see Appendix A.2). The network outputs two analog decision variables, each of which indicates evidence towards the right target (solid line) or left target (dashed line). A decision is made in the direction of whichever decision variable passes a preset threshold (0.6) first. The time at which the input passes the threshold is defined to be the reaction time.

of all artificial units and their recurrent connectivity are fully observed. It is therefore possible to engineer [22] and reverse engineer [23] RNNs by analyzing their activity and recurrent connectivity, providing mechanistic insight into cortical computation [3, 12, 21].

Traditionally, RNNs have provided insight into local computations, and there has been limited insight into multi-area computation [7, 19]. To study multi-area computation, we explicitly constrained RNNs to have multiple recurrent areas, which we refer to as multi-area RNNs. We used these multi-area RNNs to study decision-making, a cognitive process known to involve multiple areas including the prefrontal, parietal, and premotor cortex [7, 9, 12, 24-28]. Multi-area RNNs enable us to investigate several questions. Most broadly, what are the roles of within-area dynamics and inter-area connections in mediating distributed computations? How does the dimensionality and dynamics of neural computation differ across areas? What role do inter-area feedforward and feedback connections play in propagating information and rejecting noise? How do intra-area dynamics and inter-area connections coordinate to solve a task?

We use multi-area RNNs to study these questions in a decision-making task where premotor cortex and upstream areas are known to perform distinct computations. We trained multi-area RNNs to perform a perceptual decision-making task (Checkerboard Task) and compared their activity to monkey neuron recordings from the dorsal premotor cortex (PMd). We found that, when incorporating Dale's law and anatomically-informed levels of feedforward inhibition into training, PMd-resembling dynamics emerged in multi-area RNNs. Specifically, the multi-area RNN's output area (1) resembled PMd in single unit statistics and neural population activity, and (2) only retained the "output relevant" signals. Inter-area connections preferentially propagated these output relevant signals while attenuating output irrelevant signals. Our models and analyses provide a framework for studying distributed computations involving multiple areas in neural systems.

## 2 Motivation: Decision-making involves multiple brain areas

### 2.1 Checkerboard Task

In the "Checkerboard Task" [24, 26], shown in Fig. 1, a monkey was first shown left and right targets whose color (red and green) was random on each trial. The monkey was subsequently shown a central static checkerboard composed of red and green squares. The monkey was trained to discriminate the dominant color of the static checkerboard and reach to the target matching the dominant color. Since the target colors were random on each trial, this task separates the reach direction decision from the color decision [29]. This task enables studying how information related to the selection of the color of the target and information related to the direction of the reach is represented.

### 2.2 PMd Data during Checkerboard Task

We analyzed the activity of neurons from the dorsal premotor cortex (PMd), an area associated with somatomotor decisions, in monkeys performing the Checkerboard Task [30]. Neural activity

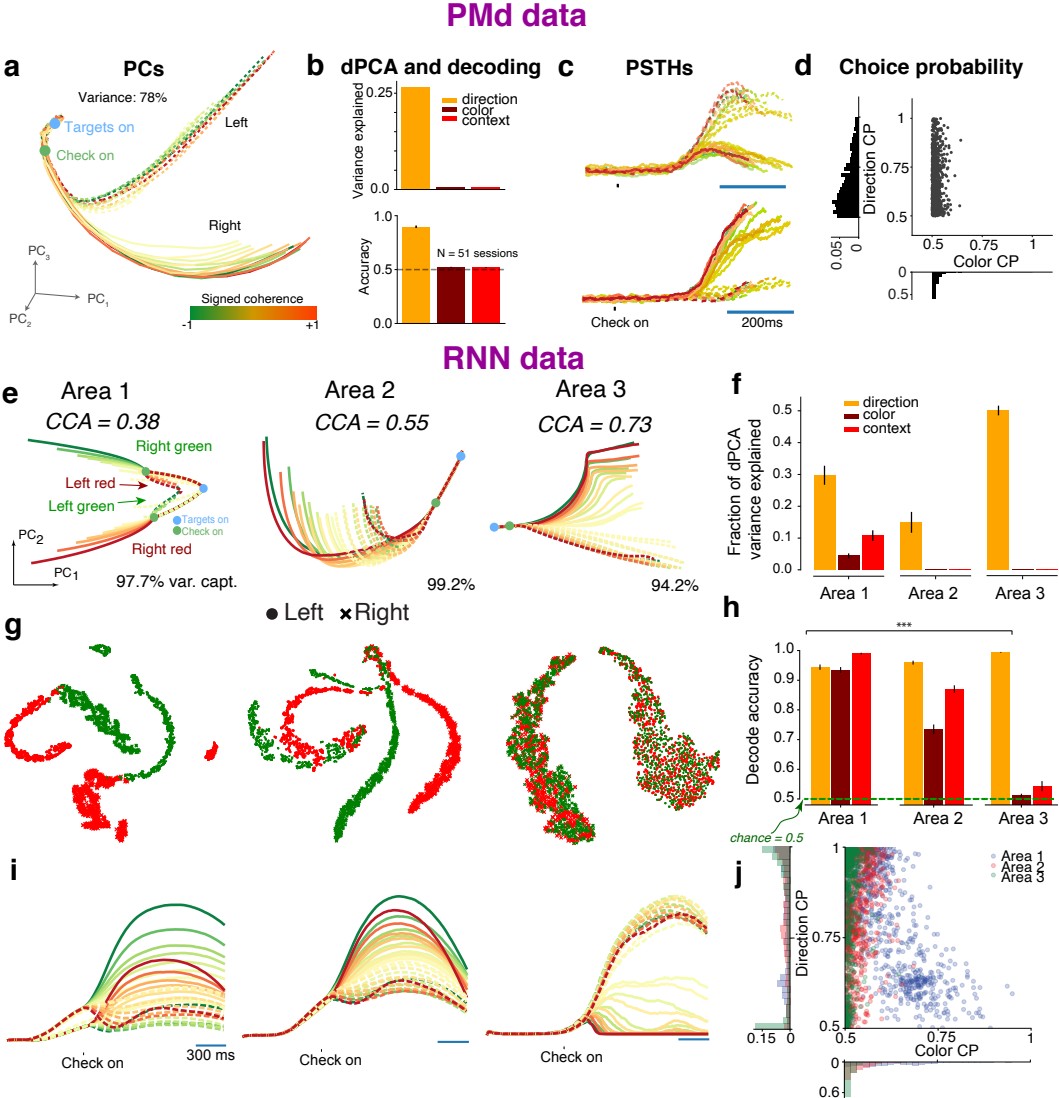

Figure 2: **PMd-resembling dynamics emerge in neuroscience constrained RNNs. (a)** PMd neural trajectories in the top 3 PCs. Color reflects signed color coherence, with darker shades of red (green) indicating more red (green) checkerboards. Right (left) reaches are denoted by solid (dotted) lines. **(b)** (Top) Variance captured by dPCA axes for the color decision, target configuration (context), and direction decision. (Bottom) Decode accuracy of the direction decision, color decision, and context in PMd sessions with U-probes and multiple neurons. **(c)** Representative PMd PSTHs aligned to checkerboard onset. **(d)** Direction and color choice probability (CP) for all recorded PMd units. **(e)** Neural trajectories in the top 2 principal components for each RNN area. **(f)** Variance captured by dPCA axes for color, context, and direction. **(g)** Non-linear tSNE embedding of peri-movement activity in each area. Each dot is a trial, with red or green denoting the color decision and '.' or 'x' denoting the direction decision. Unlike Areas 1 and 2, Area 3 only had two clusters separated based on the direction decision. **(h)** Decode accuracy of direction, color, and context in all three areas. **(i)** Example PSTHs in each area. **(j)** Choice probabilities for units in all areas (pooled over 8 RNNs).

in PMd principally reflects the direction decision (left or right) and has minimal representations associated with the dominant color of the checkerboard (red or green) [26, 30, 31]. To summarize this phenomenon, we show the principal components (PCs) of the PMd population activity in Fig. 2a. These PC trajectories separate based on the eventual reach direction (right reaches in solid, left in dotted), but not the color (red and green largely overlapping). We identified principal axes via demixed PCA (dPCA [32]) that maximized variance related to the target configuration

(context), color decision, and direction decision (see Appendix B.4). The direction axes captured significant variance (26.7%) while the color and context axes captured minimal variance (0.7%, 0.5%, respectively), as shown in Fig. 2e. It is possible, however, that there is direction-dependent color variance that is averaged away during marginalization when computing the dPCA variance [32]. Given simultaneously recorded data, a more appropriate measure of representation is the decode accuracy of direction, color and context. Across sessions where we analyzed multiple simultaneously recorded units from U-probes, the direction decision could be decoded from PMd activity significantly above chance (accuracy: 0.89, $p < 0.01$, bootstrap), but the color decision and context decode accuracy were not significantly above chance in any session (overall accuracies: 0.52 and 0.52, respectively, Fig. 2b, bottom).

Single neurons also had minimal color separation in individual PSTHs (e.g., Fig. 2c). To summarize this effect in single neurons, we computed the choice probabilities (CPs) reflecting how well the direction decision (direction CP) and color decision (color CP) could be decoded. PMd units generally had near chance color CP (0.5), but moderate to high direction CP, as shown in Fig. 2d. Together, these results demonstrate that PMd largely represents direction-related signals, but not the color decision or target configuration context. Since PMd activity minimally represents the color of the checkerboard or the target configuration, we reasoned that checkerboard and target inputs are transformed into a direction signal upstream of PMd and that multiple brain areas are necessary for solving this task. Brain areas, including the dorsolateral prefrontal cortex (DLPFC), and the ventrolateral prefrontal cortex (VLPFC), have been implicated in related sensorimotor transformations [4, 9, 33–35].

## 3 Multi-Area RNN Training Details

We trained RNNs of the form

$$\tau \dot{\mathbf{x}}(t) = -\mathbf{x}(t) + \mathbf{W}_{\mathrm{rec}}\mathbf{r}(t) + \mathbf{W}_{\mathrm{in}}\mathbf{u}(t) + \mathbf{b}_{\mathrm{rec}} + \epsilon_t, \tag{1}$$

where $\mathbf{r}(t) = \mathrm{relu}(\mathbf{x}(t))$, $\tau$ is a time-constant of the network, $\mathbf{W}_{\mathrm{rec}} \in \mathbb{R}^{N \times N}$ defines how the artificial neurons are recurrently connected, $\mathbf{b}_{\mathrm{rec}} \in \mathbb{R}^N$ defines a constant bias, $\mathbf{W}_{\mathrm{in}} \in \mathbb{R}^{N \times N_{in}}$ maps the RNN's inputs onto each artificial neuron, and $\epsilon_t$ is the recurrent noise. The output of the network is given by a linear readout of the network rates, i.e.,

$$\mathbf{z}(t) = \mathbf{W}_{\mathrm{out}}\mathbf{r}(t), \tag{2}$$

where $\mathbf{W}_{\mathrm{out}} \in \mathbb{R}^{N_{\mathrm{out}} \times N}$ maps the network rates onto the network outputs. For a 3-area RNN, $\mathbf{W}_{\mathrm{rec}}$ is defined through the following block matrix

$$\mathbf{W}_{\mathrm{rec}} = \begin{pmatrix} \mathbf{W}_{11} & \mathbf{W}_{21} & 0 \\ \mathbf{W}_{12} & \mathbf{W}_{22} & \mathbf{W}_{32} \\ 0 & \mathbf{W}_{23} & \mathbf{W}_{33} \end{pmatrix},$$

where $\mathbf{W}_{ii}$ refer to the recurrent connections of area $i$, and we use the convention that $\mathbf{W}_{i,i+1}$ refer to feedforward connections, and $\mathbf{W}_{i,i-1}$ refer to the feedback connections. Feedforward and feedback connections were only allowed between adjacent areas. Task inputs were defined to project onto the first area, and outputs were read out from the final area. In the rest of the text, we primarily focus on a 3-area RNN that had approximately 10% feedforward and 5% feedback connections between areas, based on projections between prefrontal and premotor cortex in a macaque atlas [36]. The network was also constrained to follow Dale's law, as in Song et al. [37]. The RNN processed the target context and checkerboard inputs to output decision variables reflecting accumulated evidence for a left and right decision (Fig. 1). Further details are discussed in Appendix A.2.

## 4 Results

Because the Checkerboard Task involves multiple brain areas, we reasoned that a single-area RNN would not resemble PMd recordings. We first trained traditional single-area RNNs to perform the Checkerboard Task. We found that these RNN representations mixed color and direction information, as summarized in Appendix Fig. 8, and therefore did not resemble PMd activity. This led us to study multi-area RNN models performing the Checkerboard Task, which turn out to accurately model PMd activity.

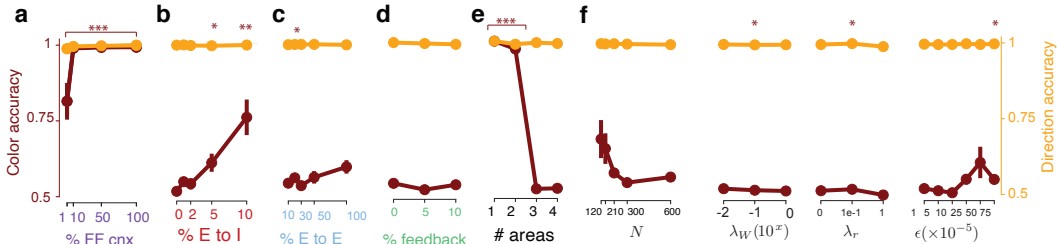

Figure 3: **PMd-resembling dynamics emerge in neuroscience constrained RNNs. (a)** We trained 3-area RNNs without explicit excitatory (E) or inhibitory (I) neurons. Inputs projected onto Area 1, and outputs were read out from Area 3. We varied the percentage of feedforward connections and computed the color and direction accuracy in Area 3. At $1\%$ feedforward connections, color could still be significantly decoded above chance. Dots are the mean across networks and error bars are s.e.m. For significance, * is $p < 0.05$, ** is $p < 0.01$, and *** is $p < 0.001$ (with appropriate correction for multiple comparisons). We incorporated Dale's law with 80% E, 20% I neurons into subsequent sweeps, **(b)** We varied the percentage of feedforward E-to-I connections. Minimal representations with chance color decode accuracy emerged when the percentage of feedforward E to I connections was $2\%$ or less (feedforward E to E was fixed at $10\%$). **(c-d)** Color information was relatively robust to feedforward E-E connections and feedback connections. **(e)** At least 3 areas were required for the RNN's last area to resemble PMd dynamics. **(f)** 3-area RNNs with neurophysiological constraints had minimal representations that were generally robust to machine learning hyperparameters. The only exceptions were when the number of units was relatively small, or the learning rate was relatively large.

## 4.1 PMd-like representations emerge in optimized multi-area RNNs with neuroscience constraints

Given the anatomical and physiological evidence suggesting that multiple brain areas are implicated in the CB task, we hypothesized that the last area of an optimized multi-area RNN would more closely resemble PMd, receiving transformed direction signals computed using the checkerboard coherence and target configuration from upstream areas. We trained multi-area RNNs to perform the Checkerboard Task as described in Section 3.

The 3-area RNN had qualitatively different population trajectories across areas, shown in Fig. 2e. Area 1 had four distinct trajectory motifs corresponding to the four possible task outcomes (combinations of left vs right and red vs green decisions). $PC_1$ primarily varied with direction, while $PC_2$ varied with both the target context and red versus green checkerboards. In contrast, Area 2 and Area 3 population trajectories primarily separated on direction, not color, like in PMd. Area 3 trajectories most strongly resembled PMd trajectories (canonical correlations, $r = 0.38, 0.55, 0.73$ for Areas 1, 2, and 3; see Appendix B.7).

We quantified the variance captured by dPCA principal axes for the context, color, and direction axis. We found that color axis variance decreased in later areas (Area 1: $5.6\%$, Area 2: $0.13\%$, Area 3: $0.07\%$, Fig. 2f). In contrast, Area 3 had the largest direction axis variance (Area 1: $30.9\%$, Area 2: $18.2\%$, Area 3: $48.5\%$, Fig. 2f). An important assumption of dPCA is that the neural activity can be decomposed as a sum of terms that depend solely on particular task variables [38]. The color variance found by dPCA indicate that color, on its own, did not account for a large fraction of the overall neural variance. However, it is possible there is significant color variance within a reach direction that dPCA, a linear dimensionality reduction technique, does not capture.

As we are interested in whether the color information is contained in the representation, a more appealing measure is decode accuracy. If the color of the target can be decoded from the representation of neural activity, then color information is present in the representation. We performed nonlinear dimensionality reduction via t-distributed stochastic neighbor embedding (tSNE) [39], shown in Fig. 2g. These results suggest that Areas 1 and 2 contain color information, but Area 3 does not (color decisions overlap). We decoded the color decision and context (target configuration) from RNN units in each area (Fig. 2h, Area 1, 2, and 3 color accuracy: 0.93, 0.76, 0.51, and Area 1, 2, and 3 context accuracy: 0.99, 0.87, 0.54). Area 1 and 2 had above chance context and color decode

accuracies ($p < 0.01/9$, 1-tailed t-test with Bonferroni correction), while Area 3 color and context decode accuracies were near chance, and most similar to PMd (Fig. 2h, color: $p = 0.05$, context: $p = 0.024$). The direction decision could be decoded significantly above chance in all areas (Fig. 2h, $p < 0.01/9$). We also observed that Area 3 unit PSTHs more closely resembled PMd neuron PSTHs (e.g., Fig. 2i), and color CP progressively decreased in later areas (Fig. 2j). Area 3, like PMd, had many neurons with moderate to high direction CP, but low color CP.

We tested how robust these results were to architecture and hyperparameter selection[2]. In particular, we quantified how well color could be decoded in the multi-area RNN's last area across several hyperparameter settings. We found that architecture impacted whether optimized multi-area RNNs had PMd-like minimal representations. In particular, we found that PMd-like representations emerged when we incorporated anatomical and neurophysiological constraints: Dale's law, empirical levels of feedforward inhibition, and at least 3 areas (Fig. 3a-e). When we varied machine learning hyperparameters, we found that our results were generally robust: multi-area RNNs had PMd-like representations in their last area over a wide range of hyperparameter settings (Fig. 3f). Together, this constellation of results shows that Area 3 of the multi-area RNN recapitulates key features of PMd activity, making this RNN a candidate model of multi-area decision-making in the Checkerboard Task.

In the next sections, we leverage the full observability of this biologically-plausible multi-area RNN to understand the mechanisms in different areas of the network and also how the network filters color information while propagating direction information.

## 4.2 Separation of the color and direction decision in Area 1

What are the key computational features of how the multi-area RNN represents color and direction information in the Checkerboard task? We first focused our analysis on Area 1, which uniquely has substantial variance for both color and direction decisions (Fig. 2h), implying a central role in computing the direction choice. We performed dPCA to identify demixed principal components related to the RNN inputs (coherence and context) and decisions (color and direction) [32]. We found demixed components that separated information related to coherence, context, the color choice, and the direction choice (Fig. 4b), consistent with these quantities being decodable from activity (Fig. 2h). We subsequently identified the context, color, and direction axes as the dPCA principal axes (unit norm, analogous to PCA eigenvectors), which combine the demixed components (analogous to PCA scores) to reconstruct neural activity [32].

We projected RNN activity and input representations onto the principal axes for context, color, and direction (Fig. 4a). We found that the context and color axis both responded to context and color inputs, and overall trajectories represented both context and color information. This suggests that color and context information are mixed in Area 1. In contrast, the direction axis strongly represented the direction choice, but did not strongly represent context or color (Fig. 4a, right). Strikingly, context and color inputs had nearly zero projection on the direction axis (Fig. 4a, right, opaque traces at 0). Consistent with these observations, we found the color and context axes were highly overlapping (dot product: 0.93), indicating that context and checkerboard variance are mixed in Area 1 activity. In contrast, the direction axes was closer to orthogonal to the context and color axes (overlap with color and context: 0.14 and 0.09, respectively).

These conclusions were upheld when we performed targeted dimensionality reduction (TDR), where we found (1) a direction axis separating left and right choices, with negligible input projections, and (2) that color and context representations were mixed (Appendix Fig. 14). Further, this structure was unique for PMd-like 3-area RNNs. In single-area RNNs, dPCA identified nearly orthogonal context, color, and direction axes, with trajectories that separated almost exclusively based on context, color, and direction, respectively (Appendix Fig. 13a).

This Area 1 representation has an important property: the direction choice is represented robustly on a nearly orthogonal axis that has close to zero context and color input projections, (Fig. 4a). This is not trivial: as counter-examples, single-area RNNs use direction axes that have context and color input projections (Fig. 13a), while the direction axis of an unconstrained 3-area RNN (without anatomical

---

[2]These sweeps over different random initializations and different parameter settings consisted of the most significant computational cost, roughly requiring 500 CPU hours on AWS, with each model training in approximately 1-2 hours.

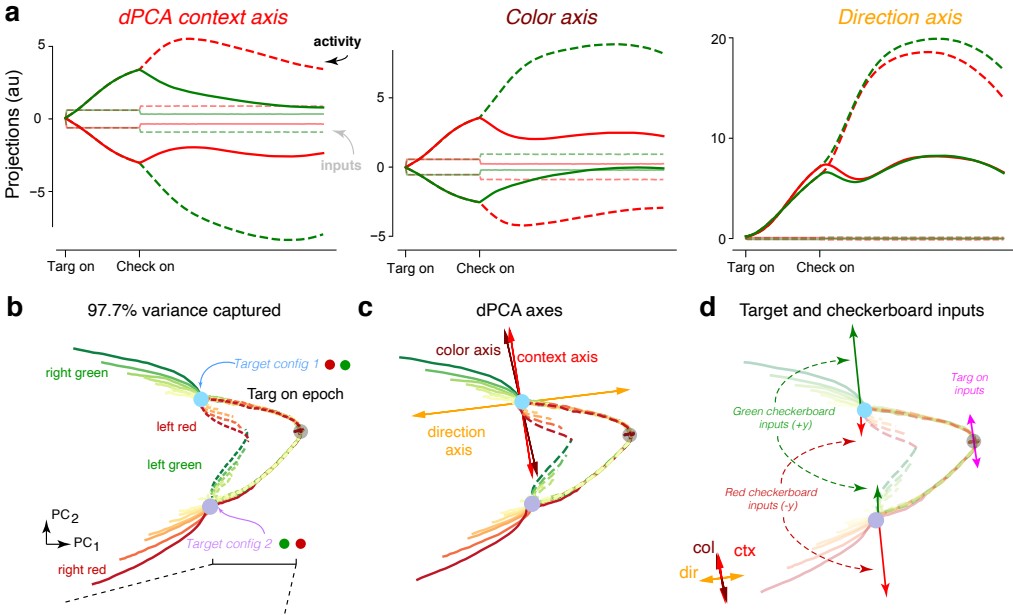

Figure 4: **Separation of direction and color in Area 1. (a)** The context, color, and direction axis correspond to the dPCA principal axes, which are not constrained to be orthogonal. Trajectories for different contexts and colors were separable on both the context and color axis. In contrast, the direction axis separated primarily on chosen direction. The RNN input representation had strong projections on the context and color axes, but not the direction axis. **(b)** Top 2 PCs of Area 1 activity, which capture 97.7% of the Area 1 variance. In the targets on epoch, the trajectories separate to two regions corresponding to the two potential target configurations (Target config 1 in blue, and Target config 2 in purple). The trajectories separate upon checkerboard color input, leading to four total trajectory motifs: right green, left red, right red, and left green. **(c)** Projection of the dPCA principal axes onto the PCs. **(d)** Projection of the context and color inputs onto the PCs. Context inputs are shown in pink, a green checkerboard input in green, and a red checkerboard input in red. Green (red) checkerboards lead to an increase (decrease) in $PC_2$ and the color axis, and differ in magnitude depending on the location of the trajectory in PC space. Trajectories are reduced in opacity to better visualize inputs.

connectivity constraints that did not resemble PMd) has context and color information, and also receives context and color inputs (Fig. 13b). We show the axes overlapped with the PCs in Fig.4, as well as the effect of the checkerboard and target inputs, which qualitatively shows that the inputs do not project onto the direction axis.

### 4.3 Inter-area connections preferentially propagate output-relevant direction information

The differentiating aspect of multi-area computation is that the different areas are separated. A natural question to ask is how then does information propagate between areas? As defined in Section 3, we denote the feedforward connections from Area 1 to 2 as $\mathbf{W}_{12}$, and from Area 2 to 3 as $\mathbf{W}_{23}$. We present results for feedforward connections from excitatory connections to excitatory units. Based on the hypothesis that the brain uses null and potent spaces to selectively filter information [40], we evaluated the effective potent and null spaces of $\mathbf{W}_{12}$ and $\mathbf{W}_{23}$. We defined the effective potent space to be the right singular vectors corresponding to the largest singular values (see Appendix B.9). The effective null space corresponded to the singular vectors with the smallest singular values.

We quantified how the color and direction axis were aligned with these potent and null spaces (see Appendix B.9). The projections onto the potent space are shown in Fig. 5a,b for $\mathbf{W}_{12}$ and $\mathbf{W}_{23}$, respectively. The null projection magnitudes are equal to one minus the potent projection. We found the direction axis was more aligned with the potent space and the color axis was more aligned with the null space. In fact, the direction axis (computed using the activity in Area 1) was consistently

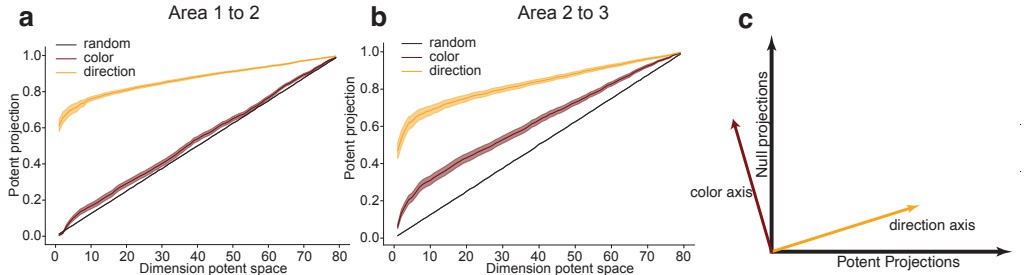

Figure 5: **(a)** Projections onto the potent space between Areas 1 and 2 for the color and direction axis, and a random vector as a function of effective rank for the input area to the middle area. Regardless of the dimension of the potent space, the direction axis is preferentially aligned with the potent space, indicating the information along this axis propagates, while the color axis is approximately randomly aligned. Shading indicates s.e.m. **(b)** Same as (a) but for projections between Areas 2 and 3. **(c)** Illustration depicting how the orientation of the axes affect the information that propagates.

most aligned with the top singular vector (governed by the parameters of the feedforward matrix; which do not affect the activity in Area 1). In contrast, the color axis was similarly aligned to a random vector. This alignment was robust to the dimension of the effective potent space, and was consistent across networks with varying feedforward connectivity percentages (10%, 20%, 30%, 50%, 100%). This suggests that learning in the multi-area recurrent network involved aligning the relevant information (in the activations) with the top singular vector (governed by the learned parameters of the feedforward matrix). These results indicate that direction information is preferentially propagated to subsequent areas, while color information is not. This phenomena is schematized in Fig. 5c. To better understand the propagation and filtering of information in networks that had color information in the output area, we performed the same analyses on networks trained without Dale's law and 2 area networks, and found that these networks had significantly reduced alignment of the direction axis with the top singular vectors (Appendix Fig. 17).

These results also have implications on how inter-area connections relay information between areas. Color activity has significant representation in Area 1 (see Fig. 2). Therefore, the inter-area connections must not merely propagate the highest variance dimensions of a preceding area [41]. Consistent with this reasoning, we found that while the top 2 PCs capture 97.7% excitatory unit variance, the top 2 readout dimensions of $\mathbf{W}_{12}$ only captured 40.0% of Area 1's excitatory unit neural variance (Appendix Fig. 16). Hence, inter-area connections are not aligned with the most variable dimensions, but are rather aligned to preferentially propagate certain types of information — a result consistent with a recent study analyzing links between activity in V1 and V2 [41].

### 4.4 Area 3, modeling PMd dynamics, is primarily input driven and implements bistable dynamics

We showed previously that Area 3 most closely resembled PMd's dynamics (Fig. 2). Our results suggest that a direction signal has been computed before Area 3 and is selectively propagated through the RNN's inter-area connections. We found that the input to Area 3 (through $\mathbf{W}_{23}$) is a graded direction signal that provides a directional evidence signal for left or right reaches (Fig. 6a). This activity must be transformed into eventual DV outputs, which are the accumulated evidence for a left or right reach. This is illustrated in Fig. 6a, where we plot $\mathbf{W}_{23}\mathbf{r}_t^2$ ($\mathbf{r}_t^2$ are the unit activations of Area 2), and $\mathbf{r}_t^3$.

To analyze Area 3's dynamics, we first observed that $\mathbf{W}_{out}$'s coefficients were sparse, with 44 out of 80 output weights being identically zero. We found that the readout led to two separate clusters of artificial units: units with non-zero coefficients for the left DV (orange) and those with non-zero coefficients for the right DV (blue). Artificial units projected either to the left or right DV outputs, but not both, suggesting that there are two clusters mediating left and right choices.

Based on this clustering, we sorted and visualized the connections of excitatory units of Area 3, which upon first glance generally has no discernible structure (Fig. 6c, left panel). After sorting, we found that two self-excitatory pools of units emerged in $\mathbf{W}_{rec}$, the first pool in Fig. 6c (right)

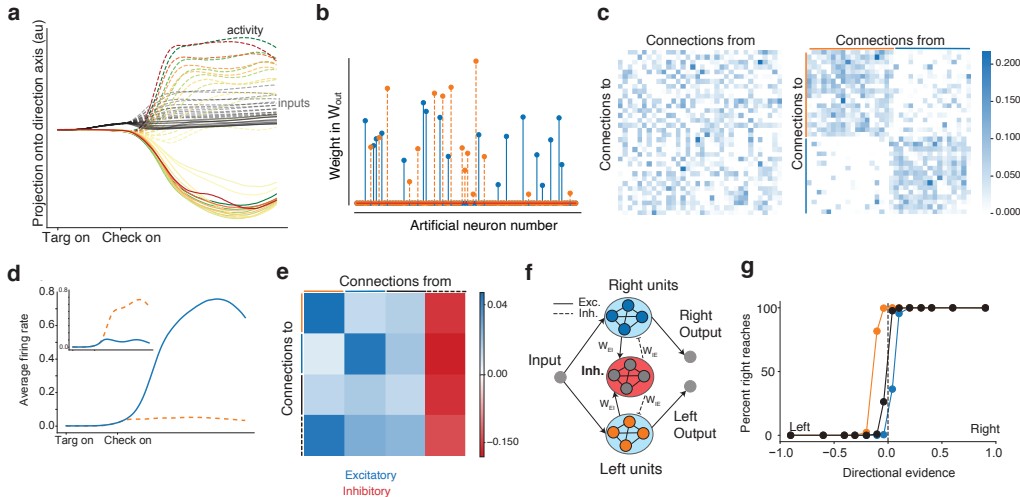

Figure 6: **Area 3 mechanism. (a)** Projection of input and overall activity onto the direction axis identified through dPCA. The conventions are the same as in Fig. 4. **(b)** Readout weights in $\mathbf{W}_{\text{out}}$ are sparse, with many zero entries, and selective weights for a left or right reach. **(c)** The unsorted connectivity matrix for the nonzero readout units (left panel), and the sorted connectivity matrix when the matrix was reordered based on the readout weight pools (right). **(d)** Average PSTHs from units for a leftward reach and (inset) rightwards reach. When one pool increases activity, the other pool decreases activity. **(e)** Averaged recurrent connectivity matrix. **(f)** Schematic of output area. **(g)** Psychometric curve after perturbation experiment, where $10\%$ of inhibitory weights to the left pool (orange) and right pool (blue) were increased (doubled). Directional evidence is computed by using the signed coherence and using target configuration to identify the strength of evidence for a left reach and strength of evidence for a right reach. Increasing inhibition to the left excitatory pool leads to more right choices and vice versa.

corresponding to the left DV and the second pool corresponding to the right DV. In addition to these two pools, we identified a pool of randomly connected excitatory units and a pool of inhibitory units with strong projections from and to the two pools. The full Area 3 connectivity matrix is shown in Appendix Fig. 18. This structure is consistent with a winner-take-all network, where increasing activity in one pool inhibits activity in the other pool through a separate inhibition pool (Fig. 6d). By taking the averaged connectivity matrix, similar to [42], we confirmed that there were two excitatory pools that received similar projections from the random excitatory pool and inhibitory pool (Fig. 6e). We summarize the behavior with a schematic of the area in Fig. 6f.

We subsequently applied selective perturbations to $\mathbf{W}_{\text{rec}}$ to determine how behavioral performance was biased. We increased inhibition to either the right or the left pool by doubling the weights of 10% of the inhibitory neurons associated with each pool. We found that this biased the network towards more left or right reaches, respectively, shown in Fig. 6g. When inhibition was increased to the right excitatory pool, the network was more likely to respond left. Conversely, when inhibition was increased to the left excitatory pool, the network was more likely to respond right.

Together, these results show that the output area, modeling PMd, robustly transforms separable direction inputs to a decision variable through a winner-take-all mechanism.

## 5    Discussion

Even though behavior and cognition arise from the coordinated computations of multiple brain areas, there is limited understanding of how interacting brain areas coordinate to produce cognitive behavior [41, 43]. In this study, we used multi-area RNNs to gain mechanistic insight into how the brain computes a perceptual decision in the Checkerboard Task and transmits only the direction decision to PMd. These results propose hypotheses for computations that occur upstream of PMd, particularly how neural population activity representing context, color, and direction are structured, and what

information is propagated between areas. We found that inter-area connections were preferentially aligned to the direction axis, not axes of maximal variance, leading to selective propagation of direction activity and attenuation of color activity. This role for inter-area connections is consistent with null and potent spaces for filtering and propagating information between areas [40, 44] and communication subspaces, which are aligned with lower variance dimensions [41].

Our results suggest that cortex and multi-area RNNs may share a more general principle of multi-area information processing: if information becomes irrelevant for later computations, it is reduced or discarded. In the Checkerboard Task, color information is necessary to compute the direction decision, but does not need to be represented after the direction decision is computed, as in PMd [4, 26, 30, 31, 45]. In deep neural networks, it is believed that minimal representations simplify the role of the output classifier [46, 47]. This idea is consistent with (1) the multi-area RNN developing a minimal (little color information) but sufficient (robust direction information) representation of task inputs, and (2) Area 3, the output area, using a simple winner-take-all readout, forming two pools of neurons representing right and left decisions (Fig. 6).

Our analysis of the multi-area RNN leads to testable hypotheses for future experiments. First, we expect that neurons in cortical areas upstream of PMd should exhibit mixed selectivity for color and direction information, consistent with studies of dorsolateral prefrontal cortex (DLPFC) and ventrolateral prefrontal cortex (VLPFC) in cognitive tasks [48–50]. More specifically, our model predicts the following organization of population dynamics in these areas: neural population dynamics should diverge to two regions with slow dynamics based on target configuration, with largely overlapping context and color axes, but an orthogonal direction axis. Second, due to alignment of inter-area connections, direction axis activity in DLPFC/VLPFC should be more predictive of activity in downstream regions such as PMdr and PMd than activity in the top PCs.

## Acknowledgments and Disclosure of Funding

We thank Laura Driscoll for helpful comments on the manuscript as well as Krishna V. Shenoy and William T. Newsome for helpful discussions on earlier versions of these results. We also thank Krishna V. Shenoy for kindly allowing us to use the data collected by Dr. Chandrasekaran when he was a postdoc in the Shenoy Lab. MK was supported by the National Sciences and Engineering Research Council (NSERC). CC was supported by a NIH/NINDS R00 award R00NS092972 and R01 award NS122969, the Moorman-Simon Interdisciplinary Career Development Professorship from Boston University, the Whitehall foundation, and the Young Investigator Award from the Brain and Behavior Research Foundation. JCK was supported by NSF CAREER 1943467, NIH DP2NS122037, the Hellman Foundation, and a UCLA Computational Medicine AWS grant. We gratefully acknowledge the support of NVIDIA Corporation with the donation of the Titan Xp GPU used for this research.

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
