# Appendix

## Table of Contents

# A   Task and training details

## A.1   Somatomotor reaction time visual discrimination task and recordings from PMd:

The task, training and electrophysiological methods used to collect the data used here have been described previously [30] and are reviewed briefly below. All surgical and animal care procedures were performed in accordance with National Institutes of Health guidelines and were approved by the Stanford University Institutional Animal Care and Use Committee. Two trained monkeys (Ti and Ol) performed a visual reaction time discrimination task. The monkeys were trained to discriminate the dominant color in a central static checkerboard composed of red and green squares and report their decision with an arm movement. If the monkey correctly reached to and touched the target that matched the dominant color in the checkerboard, they were rewarded with a drop of juice. This task is a reaction time task, so that monkeys initiated their action as soon as they felt they had sufficient evidence to make a decision. On a trial-by-trial basis, we varied the signed color coherence of the checkerboard, defined as $(R - G)/(R + G)$, where R is the number of red squares and G the number of green squares. The color coherence value for each trial was chosen uniformly at random from 14 different values arranged symmetrically from 90% red to 90% green. Reach targets were located to the left and right of the checkerboard. The target configuration (left red, right green; or left green, right red) was randomly selected on each trial. Both monkeys demonstrated qualitatively similar psychometric and reaction-time behavior. 996 units were recorded from Ti (n=546) and Ol (n=450) while they performed the task [30]. Monkey Ol and Ti's PMd units both had low choice color probability. Reported analyses from PMd data use units pooled across Monkey Ol and Ti.

## A.2   RNN description and training

We trained a continuous-time RNN to perform the checkerboard task. The RNN is composed of $N$ artificial neurons (or units) that receive input from $N_{\text{in}}$ time-varying inputs $\mathbf{u}(t)$ and produce $N_{\text{out}}$ time-varying outputs $\mathbf{z}(t)$. The RNN defines a network state, denoted by $\mathbf{x}(t) \in \mathbb{R}^N$; the $i$th element of $\mathbf{x}(t)$ is a scalar describing the "currents" of the $i$th artificial neuron. The network state is transformed into the artificial neuron firing rates (or network rates) through the transformation:

$$\mathbf{r}(t) = f(\mathbf{x}(t)), \tag{3}$$

where $f(\cdot)$ is an activation function applied elementwise to $\mathbf{x}(t)$. The activation function is typically nonlinear, endowing the RNN with nonlinear dynamics and expressive modeling capacity [51]. In this work, we use $f(x) = \max(x, 0)$, also known as the rectified linear unit, i.e., $f(x) = \text{relu}(x)$. In the absence of noise, the continuous time RNN is described by the equation

$$\tau \dot{\mathbf{x}}(t) = -\mathbf{x}(t) + \mathbf{W}_{\text{rec}}\mathbf{r}(t) + \mathbf{W}_{\text{in}}\mathbf{u}(t) + \mathbf{b}_{\text{rec}} + \epsilon_t, \tag{4}$$

where $\tau$ is a time-constant of the network, $\mathbf{W}_{\text{rec}} \in \mathbb{R}^{N \times N}$ defines how the artificial neurons are recurrently connected, $\mathbf{b}_{\text{rec}} \in \mathbb{R}^N$ defines a constant bias, $\mathbf{W}_{\text{in}} \in \mathbb{R}^{N \times N_{in}}$ maps the RNN's inputs onto each artificial neuron, and $\epsilon_t$ is the recurrent noise. The output of the network is given by a linear readout of the network rates, i.e.,

$$\mathbf{z}(t) = \mathbf{W}_{\text{out}}\mathbf{r}(t), \tag{5}$$

where $\mathbf{W}_{\text{out}} \in \mathbb{R}^{N_{\text{out}} \times N}$ maps the network rates onto the network outputs.

We trained RNNs to perform the checkerboard task as follows. For all networks, unless we explicitly varied the amount of units, we used $N_{\text{in}} = 4$, $N = 300$, and $N_{\text{out}} = 2$.

The four inputs were defined as:

1. Whether the left target is red (-1) or green (+1).
2. Whether the right target is red (-1) or green (+1).
3. Signed coherence of red (ranging from -1 to 1), $(R - G)/(R + G)$.
4. Signed coherence of green (ranging from -1 to 1), $(G - R)/(R + G)$. Note that, prior to the addition of noise, the sum of the signed coherence of red and green is zero.

The inputs, $\mathbf{u}(t) \in \mathbb{R}^4$, were defined at each time step, $t$, in distinct epochs. In the 'Center Hold' epoch, which lasted for a time drawn from distribution $\mathcal{N}(200 \text{ ms}, 50^2 \text{ ms}^2)$, all inputs were set to

zero. Subsequently, during the 'Targets' epoch, which lasted for a time drawn from distribution $\mathcal{U}[600\text{ ms}, 1000\text{ ms}]$, the colors of the left and right target were input to the network. These inputs were noiseless, as illustrated in Fig. 1, to reflect that target information is typically unambiguous in our experiment. Following the 'Targets' epoch, the signed red and green coherences were input into the network during the 'Decision' epoch. This epoch lasted for 1500 ms. We added zero mean independent Gaussian noise to these inputs, with standard deviation equal to 5% of the range of the input, i.e., the noise was drawn from $\mathcal{N}(0, 0.1^2)$. At every time point, we drew independent noise samples and added the noise to the signed red and green coherence inputs. We added recurrent noise $\epsilon_t$, adding noise to each recurrent unit at every time point, from a distribution $\mathcal{N}(0, 0.05^2)$. Following the 'Decision' epoch, there was a 'Stimulus Off' epoch, where the inputs were all turned to 0.

The two outputs, $\mathbf{z}(t) \in \mathbb{R}^2$ were defined as:

1. Decision variable for a left reach.
2. Decision variable for a right reach.

We defined a desired output, $\mathbf{z}_{\text{des}}(t)$, which was 0 in the 'Center Hold' and 'Targets' epochs. During the 'Decision' epoch, $\mathbf{z}_{\text{des}}(t) = 1$. In the 'Stimulus Off' epoch, $\mathbf{z}_{\text{des}}(t) = 0$. In RNN training, we penalized output reconstruction using a mean-squared error loss,

$$\mathcal{L}_{\text{mse}} = \frac{1}{|\mathcal{T}|} \sum_{t \in \mathcal{T}} |\mathbf{z}(t) - \mathbf{z}_{\text{des}}(t)|^2. \tag{6}$$

The set $\mathcal{T}$ included all times from all epochs except for the first 200 ms of the 'Decision' epoch from the loss. We excluded this time to avoid penalizing the output for not immediately changing its value (i.e., stepping from 0 to 1) in the 'Decision' epoch. Decision variables are believed to reflect a gradual process consistent with non-instantaneous integration of evidence, e.g., as in drift-diffusion style models, rather than one that steps immediately to a given output.

To train the RNN, we minimized the loss function:

$$\mathcal{L} = \mathcal{L}_{\text{mse}} + \frac{\lambda_{\text{in}}}{NN_{\text{in}}} \|\mathbf{W}_{\text{in}}\|_F^2 + \frac{\lambda_{\text{rec}}}{N^2} \|\mathbf{W}_{\text{rec}}\|_F^2 + \frac{\lambda_{\text{out}}}{NN_{\text{out}}} \|\mathbf{W}_{\text{out}}\|_F^2 + \frac{\lambda_r}{T} \sum_t \|\mathbf{r}(t)\|^2 + \lambda_\Omega \mathcal{L}_\Omega, \tag{7}$$

where

- $\|\mathbf{A}\|_F$ denotes the Frobenius norm of matrix $\mathbf{A}$
- $\lambda_{\text{in}} = \lambda_{\text{rec}} = \lambda_{\text{out}} = 1, \lambda_r = 0$ to penalize larger weights.
- $\lambda_\Omega = 2$
- $\mathcal{L}_\Omega$ is a regularization term that ameliorates vanishing gradients proposed and is described in prior literature [37, 52].

During the training process, we also incorporated gradient clipping to prevent exploding gradients [52]. Training was performed using stochastic gradient descent, with gradients calculated using backpropagation through time. For gradient descent, we used the Adam optimizer, which is a first order optimizer incorporating adaptive gradients and momentum [53].

Every 200 or 500 training epochs, we generated 2800 cross-validation trials, 100 for each of the 28 possible conditions (14 coherences × 2 target configurations). For each trial, there was a correct response (left or right) based on the target configuration and checkerboard coherence. When training, we defined a "correct decision" to be when the RNNs DV for the correct response was greater than the other DV and the larger DV was greater than a pre-set threshold of 0.6. We evaluated the network 500ms before the checkerboard was turned off (the end of the trial). We required this criteria to be satisfied for at least 65% of both leftward and rightward trials. We note that this only affected how we terminated training. It had no effect on the backpropagated gradients, which depended on the mean-squared-error loss function. Note that a trial that outputted the correct target but did not reach the 0.6 threshold would not be counted towards the 65% criteria.

When testing, we defined the RNNs decision to be either: (1) whichever DV output (for left or right) first crossed a pre-set threshold of 0.6, or (2) if no DV output crossed the pre-set threshold of 0.6 by the end of the 'Decision epoch,' then the decision was for whichever DV had a higher value at

| Hyperparameter | Value |
|---|---|
| Number of units | 300 |
| Number of areas | 3 |
| Learning rate | 5e-5 |
| Time Constant | 50ms |
| Discretization bin width | 10ms |
| Rate regularization | 0 |
| Weight regularization | 1 |
| Activation function | Relu |
| Feedforward connection | 10% |
| Feedback connections | 5% |
| Dale law | Yes |

Table 1: Hyperparameters of exemplar RNN.

the end of this epoch — an approach that is well established in models of decision-making [54, 55]. If the RNN's decision on a single trial was the same as the correct response, we labeled this trial 'correct.' Otherwise, it was incorrect. The proportion of decisions determined under criterion (2) was negligible ($0.5\%$ across 100 trials for each of 28 conditions). An interpretation for criterion (2) is that if the RNN's DV has not achieved the threshold certainty level by the end of a trial, we assign the RNN's decision to be the direction for which its DV had the largest value. Finally, in training only, we introduced 'catch' trials $10\%$ of the time. On $50\%$ of catch trials, no inputs were shown to the RNN and $\mathbf{z}_{des}(t) = 0$ for all $t$. On the remaining $50\%$ of catch trials, the targets were shown to the RNN, but no coherence information was shown; likewise, $\mathbf{z}_{des}(t) = 0$ for all $t$ on these catch trials.

We trained the three-area RNNs by constraining the recurrent weight matrix $\mathbf{W}_{rec}$ to have connections between the first and second areas and the second and third areas. In a multi-area network with $N$ neurons and $m$ areas, each area had $N/m$ neurons. In our 3-area networks, each area had 100 units. Of these 100 units, 80 were excitatory and 20 were inhibitory. Excitatory units were constrained to have only positive outgoing weights, while inhibitory units were constrained to have only negative outgoing weights. We used the `pycog` repository [37] to implement these architecture constraints. The parameters for the exemplar RNN used in the paper are shown in Table 1. In our hyperparameter sweeps, we varied the hyperparameters of the exemplar RNN. For each parameter configuration, we trained 8 different networks with different random number generator seeds.

# B    Additional description of analyses

## B.1    Decoding analysis for PMd data

For PMd data, we calculated decoding accuracy using 400 ms bins. We report numbers in a window [-300ms, +100 ms] aligned to movement onset. We used the MATLAB *classify* command with 75% training and 25 % test sets. Decoding analyses were performed using 5-31 simultaneously recorded units from Plexon U-probes and the averages reported are across 51 sessions. To assess whether decoding accuracies were significant on a session by session basis, we shuffled the labels 200 times and estimated the 1st and 99th percentiles for this surrogate distribution. The decode accuracy for direction, color, and context variables for a session was judged to be significant if it lay outside this shuffled accuracy. Every session had significant direction decode, while no session had significant color and context decode accuracy.

## B.2    Decoding and Mutual information for RNNs

We used a decoder and mutual information approximation to quantify the amount of information (color, context, direction) present in the network. We trained a neural network to predict a relevant choice (for example, color) on a test set from the activity of a population of units. We used 700 trials for training, and 2100 independent trials for testing. To generate the trials for training and testing, we increased the recurrent noise to be drawn from the distribution ($\mathcal{N}(0, 0.1^2)$) to prevent overfitting. For each trial, we averaged data in a window [-300ms, +100ms] around reaction time.

We trained a neural network with 3 layers, 64 units per layer, leakyRelu activation ($\alpha$=0.2), and dropout (p=0.5), using SGD, to predict the choice given the activity of the population. We removed the leakyRelu activation for the linear network, and increased dropout (p=0.8). For both the nonlinear and linear network, we trained the neural network to minimize the cross-entropy loss. We used the same neural network from the decode to compute an approximation to mutual information, described in Supplementary Note 2.

## B.3 RNN behavior

To evaluate the RNN's psychometric curve and reaction-time behavior, we generated 200 trials for each of the 28 conditions, producing 400 trials for each signed coherence. For these trials, we calculated the proportion of red decisions by the RNN. This corresponds to all trials where the DV output for the red target first crossed the preset threshold of 0.6; or, if no DV output crossed the threshold of 0.6, if the DV corresponding to the red target exceeded that corresponding to the green target. The reaction time was defined to be the time between checkerboard onset to the first time a DV output exceeded the preset threshold of 0.6. If the DV output never exceeded a threshold of 0.6, in the reported results, we did not calculate a RT for this trial.

## B.4 dPCA

Demixed principal components analysis (dPCA) is a dimensionality reduction technique that provides a projection of the data onto task related dimensions while preserving overall variance [32]. dPCA achieves these aims by minimizing a loss function:

$$L_{dpca} = \sum_c \|\mathbf{X}_c - \mathbf{P}_c \mathbf{D}_c \mathbf{X}\|^2. \tag{8}$$

Here, $\mathbf{X}_c$ refers to data averaged over a "dPCA condition" (such as time, coherence, context, color, or direction), having the same shape as $\mathbf{X} \in \mathbb{R}^{N \times cT}$, but with the entries replaced with the condition-averaged response. The aim is to recover (per dPCA condition $c$) a $\mathbf{P}_c$ and $\mathbf{D}_c$ matrix. $\mathbf{P}_c$ is constrained to have orthonormal columns, while $\mathbf{D}_c$ is unconstrained. The number of columns of $\mathbf{P}_c$ and rows of $\mathbf{D}_c$ reflects the number of components one seeks to find per condition. We project the data onto the principal components $\mathbf{D}_c \mathbf{X}$ to observe the demixed components (Fig. 4b). The column of $\mathbf{P}_c$ reflects how much the demixed data contributes to each neuron. We use the principal axes from $\mathbf{P}_c$ to compute the axis overlap, as in Kobak et al [32]. We used axes of dimension 1 for RNNs, which were sufficient to capture most color, context, or direction variance. For the neural data, we used five components for direction, color and context since the PMd data was higher dimensional than the RNNs.

Our results were consistent if we used dPCA or TDR (Fig. 14). The top principal axis from each $\mathbf{P}_c$ are analogous to the axes found from TDR. Both methods seek to reconstruct neural activity from demixed components. To apply TDR, one explicitly parametrizes task variables (See Targeted Dimensionality Reduction (Appendix B.5)), while $\mathbf{D}_c \mathbf{X}$ serves the purpose of finding demixed components in dPCA. Overall, the choice of using dPCA or TDR to find the axes did not affect our conclusions.

For multi-area analyses, we separated the units for each area and found the task-relevant axes for this subset of units. For the inter-area analyses, we used RNNs with only excitatory connections, and therefore found the color and direction axis using only the excitatory units (Fig. 15). In all other analyses, all units were used to identify the axes. For RNN activity, we performed dPCA using activity over the entire trial. For PMd activity, we used a window of (0ms, 800ms) relative to checkerboard onset. We restricted time windows for the PMd activity because we wanted to minimize movement related variance.

## B.5 Targeted Dimensionality Reduction

Targeted dimensionality reduction (TDR) is a dimensionality reduction technique that finds low dimensional projections that have meaningful task interpretations. We applied TDR as described by the study by Mante et al. [49]. We first z-scored the firing rates of each of the 300 units across time and trials, so that the firing rates had zero mean and unit standard deviation. We then expressed this

z-scored firing rate as a function of task parameters using linear regression,

$$r_{i,t}(k) = \beta_{i,t}^1 \text{color}(k) + \beta_{i,t}^2 \text{direction}(k) + \beta_{i,t}^3 \text{context}(k). \tag{9}$$

Here, $r_{i,t}(k)$ refers to the firing rate of unit $i$ at time $t$ on trial $k$. The total number of trials is $N_{\text{trials}}$. This regression identifies coefficients $\beta_{i,t}^m$ that multiply the m$^{\text{th}}$ task parameter to explain $r_{i,t}(k)$. We defined the task parameters as follows:

- color$(k)$ was the signed coherence of the checkerboard on trial $k$, given by $(R-G)/(R+G)$.
- direction$(k)$ was $-1$ for a left decision and $+1$ for a right decision.
- context$(k)$ was the target orientation, taking on $-1$ if the green (red) target was on the left (right) and $+1$ if the green (red) target was on the right (left).

We did not fit a bias term since the rates were z-scored and therefore zero mean. For each unit, $i$, we formed a matrix $\mathbf{F}_i$ having dimensions $N_{\text{trials}} \times 3$, where each row consisted of [color$(k)$, direction$(k)$, context$(k)$]. We define $\mathbf{r}_{i,t}$ to be the rate of unit $i$ at time $t$ across all trials. We then solved for the coefficients, denoted by $\boldsymbol{\beta}_{i,t} = [\beta_{i,t}^1, \ \beta_{i,t}^2, \ \beta_{i,t}^3]^T$, using least squares,

$$\boldsymbol{\beta}_{i,t} = (\mathbf{F}_i^T \mathbf{F}_i)^{-1} \mathbf{F}_i^T \mathbf{r}_{i,t}. \tag{10}$$

Each $\boldsymbol{\beta}_{i,t}$ is therefore a $3 \times 1$ vector, and concatenating $\boldsymbol{\beta}_{i,t}$ across $t$ results in $\boldsymbol{\beta}_i$, a $3 \times T$ matrix, of which there are $N$. We then formed a tensor where each $\beta_i$ is stacked, leading to a tensor with dimensions $3 \times T \times N$. For each of the 3 task variables, we found the time $T$ where the norm of the regression coefficients, across all units, was largest. For the $m$th task variable, we denote the vector $\beta_{\max}^m \in \mathbb{R}^N$ to be a vector of coefficients that define a 1-dimensional projection of the neural population activity related to the $m$th task variable. These vectors are what we refer to as the task related axes. To orthogonalize these vectors, we performed QR decomposition on the stacked $\boldsymbol{\beta}_{\max}$ matrix $[\boldsymbol{\beta}_{\max}^1, \boldsymbol{\beta}_{\max}^2, \boldsymbol{\beta}_{\max}^3]$, which is an $N \times 3$ matrix. This decomposition finds orthogonal axes so that the axes would capture independent variance.

## B.6 Choice probability

To calculate the choice probability for a single unit, we first computed the average firing rate in a window from $[-300$ ms, $+100$ ms$]$ around the reaction time for each trial. We used the average firing rates calculated across many trials to create a firing rate distribution based on either the color decision (trials corresponding to a red or green choice) or the direction decision (trials corresponding to a left or right choice).

To compute the color choice probability, we constructed the firing rate distributions corresponding to a green choice or red choice. If these two distributions are non-overlapping, then the neuron has a color choice probability of 1; the average firing rate will either overlap with the red or green firing rate distributions, but not both. On the other hand, if the two distributions are completely overlapping, then the neuron has a color choice probability of $0.5$; knowing the firing rate of the neuron provides no information on whether it arose from the red or green firing rate distribution. When there is partial overlap between these two distributions, then firing rates where the distributions overlap are ambiguous. We computed choice probability as the area under the probability density function at locations when the two distributions did not overlap, divided by 2 (to normalize the probability). To calculate the direction choice probability, we repeated the same calculation using firing rate distributions corresponding to a left choice or right choice.

## B.7 Canonical correlation

We applied CCA to assess the similarity between neural activity and the artificial unit activity [56]. Before applying CCA, we performed principal component analysis to reduce the dimensionality of the artificial and neural activity to remove noise [56]. We reduced the dimensionality to 3 and 8 for RNNs and PMd, respectively. These dimensionalities were chosen as they captured over $88\%$ of the variance for each dataset when aligned to checkerboard. We report the average CCA correlation coefficients in Fig. 2 using times in a window of [0, 400ms] aligned to checkerboard onset for the PMd and RNN activity. The data was binned in 10ms bins.

## B.8 Analyses of inputs and activity

In order to disentangle the effects of external inputs and recurrence, in Fig. 4a, we evaluated the input contribution and overall activity. For Area 1, we defined the input contribution as $\mathbf{W}_{\text{in}}\mathbf{u}_t$, and for areas 2 and 3, we defined the input contribution as $\mathbf{W}_{12}\mathbf{r}_t^1$, and $\mathbf{W}_{23}\mathbf{r}_t^2$ respectively, where $\mathbf{r}_t^m$ denotes the activity of the units in area $m$. The activity $\mathbf{r}_t^m$ corresponds to the firing rate that experimentalists could measure, reflecting a combination of input and recurrent interactions. For constant inputs, a stable value of the activity implies there is little recurrent processing.

## B.9 Inter-Area Projection Analyses

To calculate the overlap between the color and direction axes with the potent and null spaces, we performed singular value decomposition on the inter-area connections, $\mathbf{W}_{12}$ and $\mathbf{W}_{23}$. $\mathbf{W}_{12}$ and $\mathbf{W}_{23}$ were $80 \times 80$ matrices, and were full rank. Nevertheless, they had near some zero singular values, indicating that the effective rank of the matrix was less than $80$. We defined the potent dimensions to be the top $m$ right singular vectors, while the null dimensions were the remaining $80 - m$ right singular vectors.

We performed the analyses of Fig. 5a,b by varying the potent and null dimensions, sweeping $m$ from 1 to 80. For each defined potent and null space, we calculated the axis overlap between the direction (or color) axis and the potent (or null) space by computing the L2-norm of the orthogonal projection (squared). We report the squared quantity because the expectation of the norm of a projection of a random vector onto an $m$-dimensional subspace of an $n$-dimensional space is $m/n$. We include an approximation of the expectation of the projection of a random vector in Fig. 5a,b by averaging the projection of 100 random vectors. Our results show that the direction axis was always more aligned with potent dimensions than the color axis, irrespective of the choice of $m$, and that the direction axis was preferentially aligned with the top singular vector.

## B.10 Visualization of neural activity in a low dimensional space

The activity of multiple units on a single trial is high dimensional, with dimension equal to the number of units. To visualize the activity in a lower dimensional space, dimensionality reduction techniques can be used. In addition to TDR, we also utilized Principal Components Analysis (PCA) and t-distributed Stochastic Neighbor Embedding (tSNE) to visualize neural activity in low-dimensional spaces.

PCA finds a linear low-dimensional projection of the high dimensional data that maximizes captured variance. We performed PCA on both the experimental data and RNN rates. PCA is an eigenvalue decomposition on the data covariance matrix. To calculate the covariance matrix of the data, we averaged responses across conditions. This reduces single trial variance and emphasizes variance across conditions. Firing rates were conditioned on reach direction and signed coherence. In both the experimental data and RNN rates, we had 28 conditions (14 signed coherences each for left and right reaches).

tSNE embeds high dimensional data in a low dimensional manifold that is nonlinear, enabling visualization of activity on a nonlinear manifold. The tSNE embedding maintains relative distances between data points when reducing dimensionality, meaning that points closer in high dimensional space remain closer when viewed in a low dimensional manifold. We projected our data into a two dimensional manifold. We used the default parameters from the sickit-learn implementation (https://scikit-learn.org/stable/modules/generated/sklearn.manifold.TSNE.html). The data visualized under tSNE was averaged in a window [-300ms, +100ms] around reaction time.

# C  Supplementary Figures

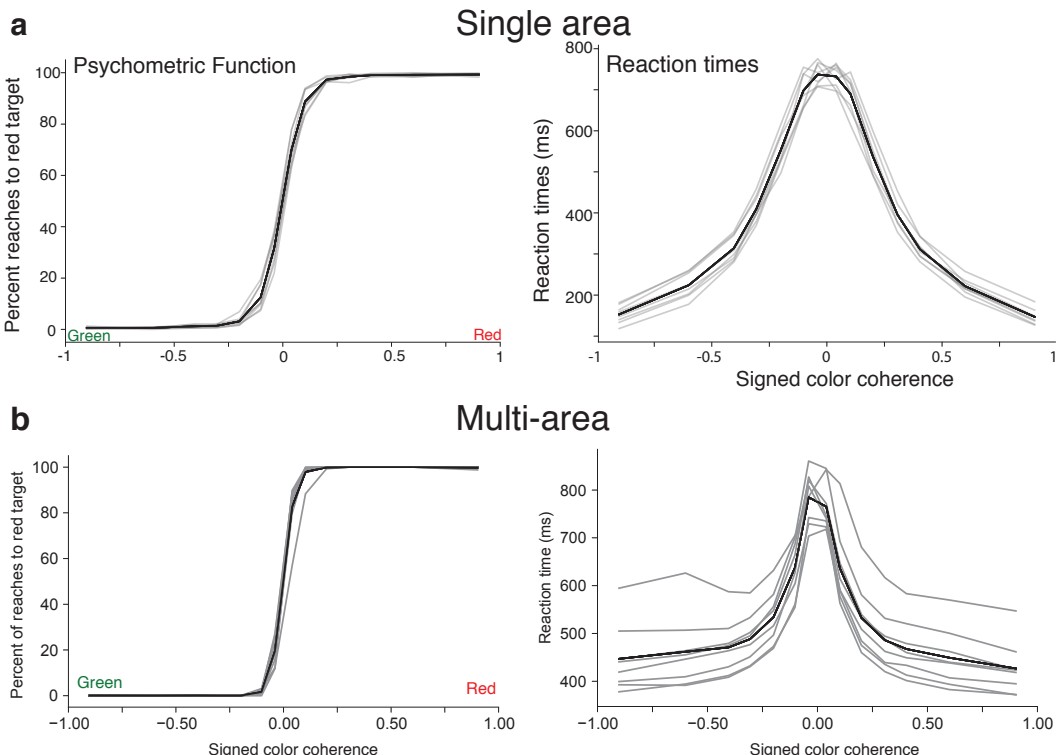

Figure 7: Psychometric and reaction time curves for single-area **(a)** and multi-area RNNs **(b)** with Dale's law trained for this study. The hyperparameters used for these RNNs are described in Table 1. Gray lines represent individual RNNs and the black solid line is the average across all RNNs.

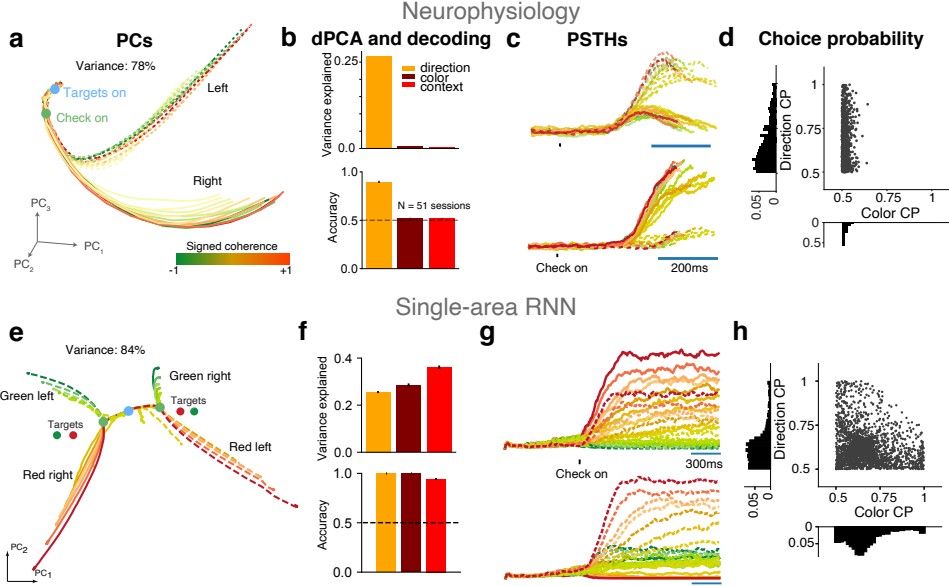

Figure 8: **Single-area RNNs do not naturally reproduce PMd dynamics. (a-d)** Reproduced from Fig. 2 for comparison. **(e)** Single-area RNN neural trajectories in the top 2 PCs. Single-area RNNs had four trajectory motifs for each combination of (left vs right) and (red vs green). In the Targets epoch, the RNN's activity approached one of two locations in state space (light green dots), corresponding to the two target configurations. In the checkerboard epoch, trajectories separate based on the coherence of the checkerboard, causing 4 total distinct trajectory motifs. Although the direction decision is not separable in the principal components, the direction decision is separable in higher dimensions (see the direction axis found using dPCA in Fig. 13a). **(f)** dPCA variance captured for the color (28%), context (26%), and direction (36%) axes for the RNN. The color and direction decisions, as well as the target configuration context, could be decoded from the RNN population activity well above chance. **(g)** Example RNN PSTHs, demonstrating coherence selectivity (top) and mixed selectivity (bottom). **(h)** Choice probability for simulated single-area RNN units. Many units have high color choice probabilities.

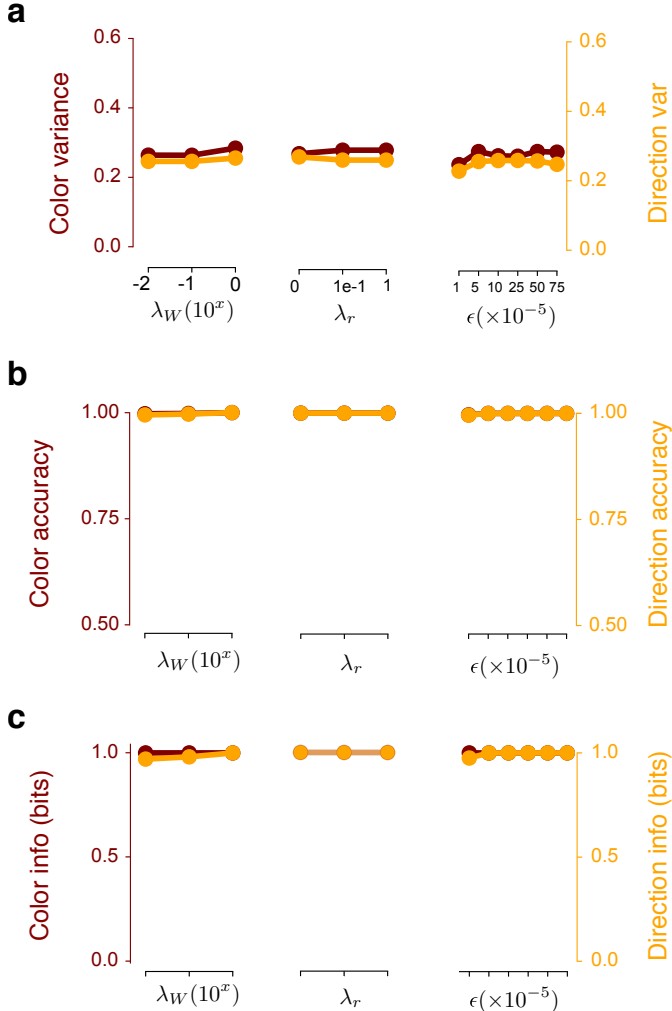

Figure 9: **Hyperparameter sweeps for single-area RNNs**. **(a)** dPCA color and direction variance captured for three different regularization parameters (weight regularization: $\lambda_w$, rate regularization: $\lambda_r$, and learning rate: $\epsilon$). There is a significant color representation in all optimized single-area RNNs. **(b)** Decode accuracy of the color and direction decision; color accuracy is at 1 (hidden behind direction accuracy) for the three different hyper parameters. The color decode accuracy (maroon) is at nearly 1 across all tested hyperparameters. These points are behind the direction decode accuracy (orange). **(c)** Mutual information estimate. The color mutual information (maroon) is nearly 1 across all tested hyperparameters.

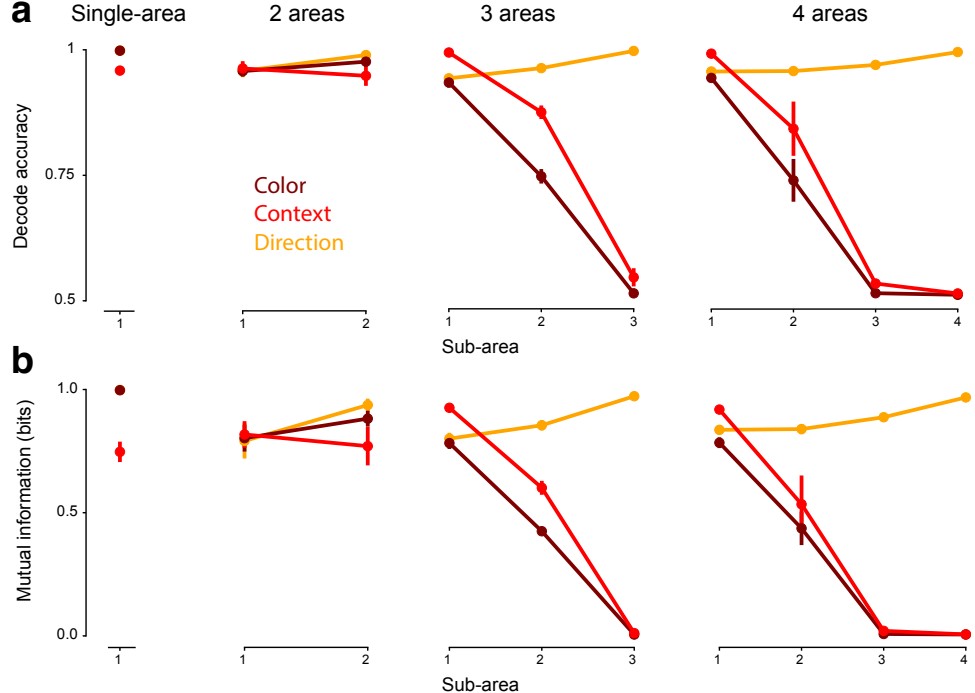

Figure 10: **Decode accuracy and mutual information per area in multi-area RNNs**. **(a)** Decode accuracy in each area for 1- to 4-area RNNs for color, context, and direction corresponding to Fig. 3d. The 3- and 4-area RNNs had minimal color representations in their last area. Note that the 4-area RNN also has a minimal color representation in Area 3. **(b)** Mutual information in each area for 1- to 4-area RNNs. Color conventions as in Fig. 3. Red is context, dark brown is color, and orange is direction.

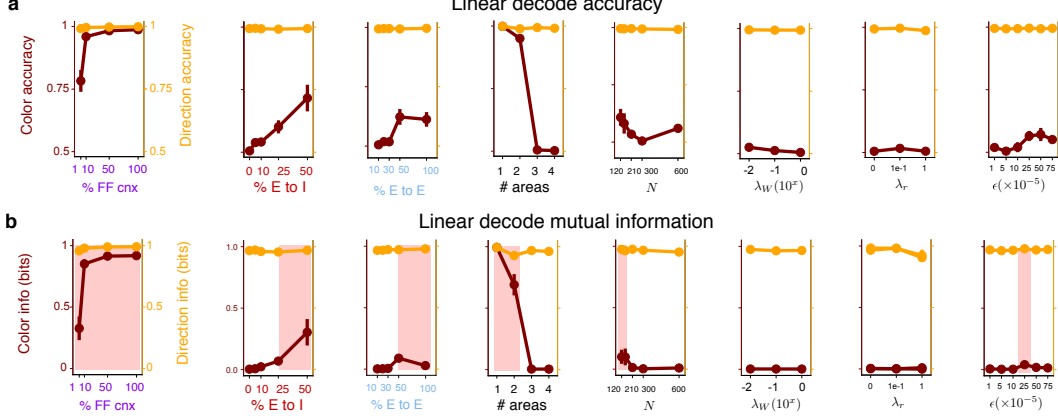

Figure 11: **Results of Fig. 3 reproduced with a linear classifier**. This figure reproduces the simulations in Fig 3, but with a linear classifier. The main conclusions are upheld. **(a)** Linear decode accuracy for all hyperparameter sweeps shown in Fig. 3. **(a)** Mutual information estimated by using the linear network trained with cross entropy loss.

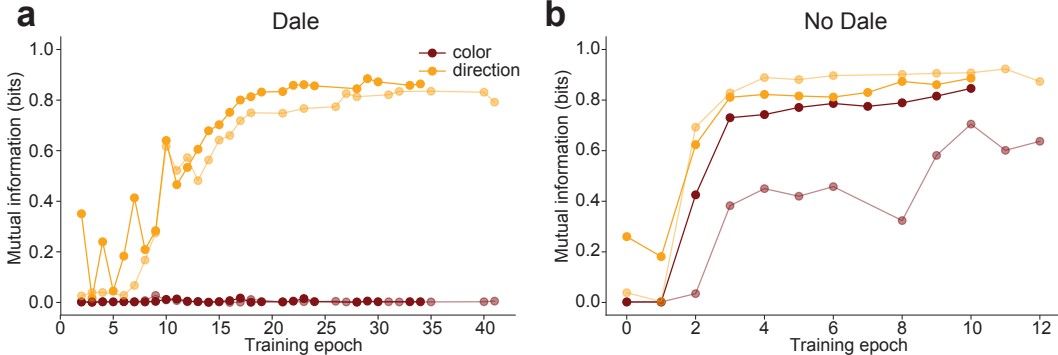

Figure 12: **Color and direction information through training in Area 3.** Each "training epoch" represents 500 iterations of gradient descent. **(a)** In the PMd-like 3-area RNNs that were trained with Dale's law, color information in Area 3 remained near zero throughout training (two different representative networks, light and dark shade). **(b)** In the unconstrained 3-area RNNs, color information in Area 3 increased early in training and appeared to plateau (two different networks, light and dark shade). Networks were only saved if the loss function decreased, so certain training epochs are not present.

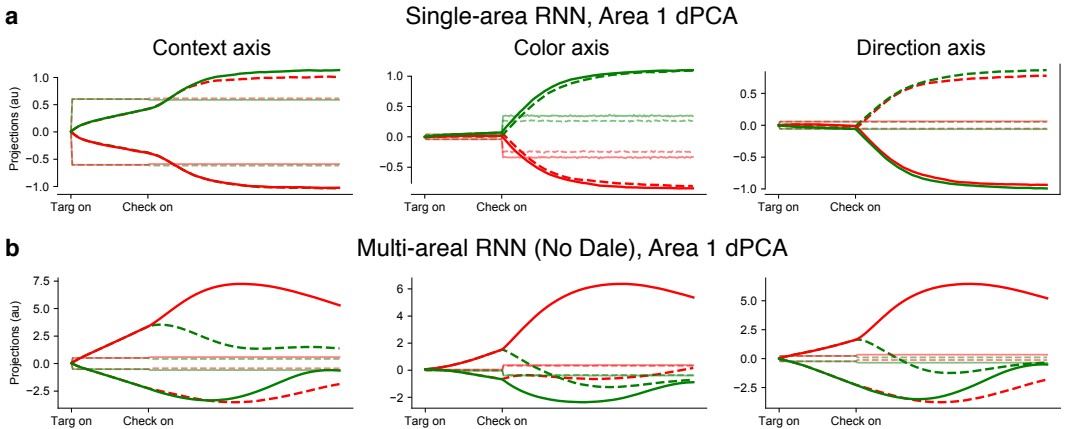

Figure 13: **dPCA trajectories for single-area and 3-area RNNs with No Dale's law. (a)** Projections onto the dPCA context, color, and direction axes for a single-area RNN. dPCA was able to find axes that separate the context input, color decision, and direction decision. Importantly, in these networks, Inputs were non-zero on the direction axis. **(b)** dPCA projections for the unconstrained 3-area RNNs with color representation in Area 3. Inputs were similarly non-zero on the direction axis. The context inputs, color decision, and direction decision, had similar projection motifs.

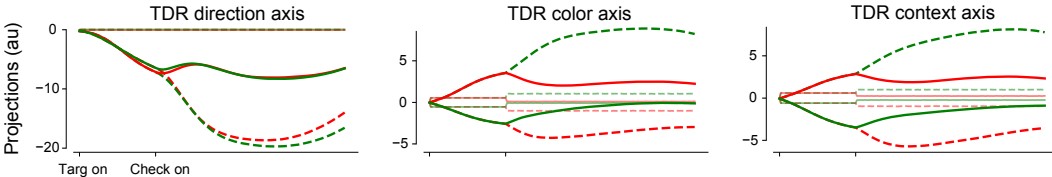

Figure 14: **TDR results closely match dPCA results, and identifies mixed color and context axes.** The direction axis separated trajectories based on the direction choice. The color and context axes had trajectory separation depending on both color and context. We did not show the orthonormalized bases, because we found that the QR decomposition was susceptible to the order in which orthonormalization was performed. This is further evidence that the color and context axes are closely aligned.

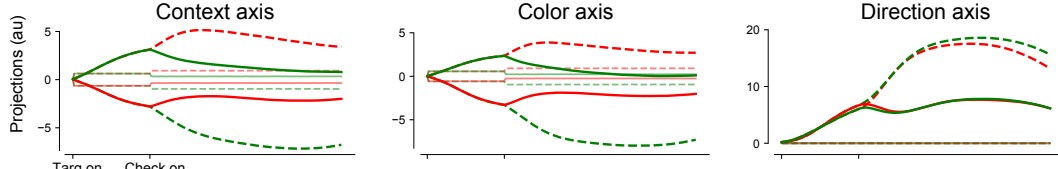

Figure 15: **dPCA projections when only considering excitatory units.** We identified the dPCA principal axes for context, color, and direction using only excitatory units. Results are consistent with the results of Fig. 4d.

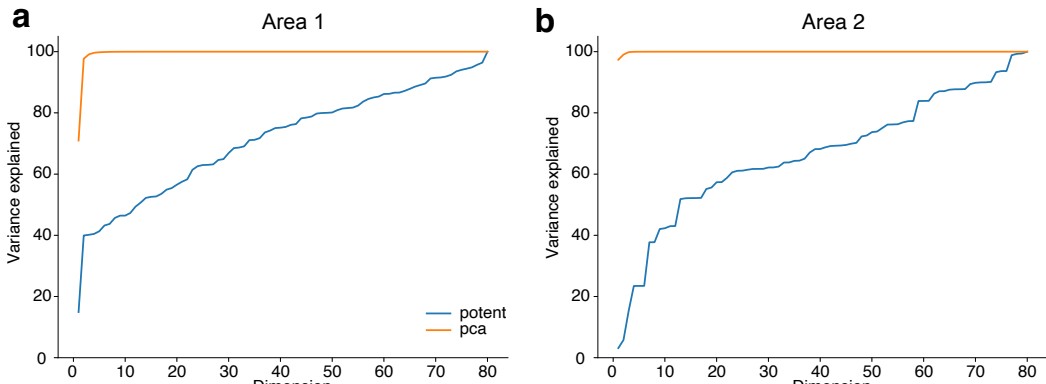

Figure 16: **Relationship between PCs and inter-area potent space. (a)** Variance explained of the excitatory units in Area 1 by the top principal components and top dimensions of potent space of $\mathbf{W}_{12}$, swept across all dimensions. **(b)** Variance explained of the excitatory units in Area 2 by the top principal components and top dimensions of potent space of $\mathbf{W}_{23}$, swept across all dimensions. These plots show that the connections between areas do not necessarily propagate the most dominant axes of variability in the source area to the downstream area. Excitatory units were used for the comparison because only excitatory units are read out by subsequent areas. These results were upheld when comparing to the variance explained by the top principal components obtained from all units.

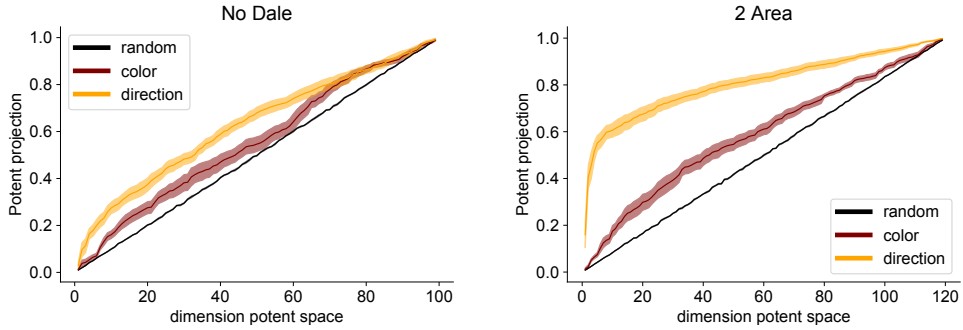

Figure 17: Projections between Area 1 and Area 2 for a network without Dale's law **(left)** and a 2 area network **(right)**, averaged across 8 trained networks. The conventions are the same as in Fig. 5. The alignment of the direction axis with the top singular vectors is reduced (compare to Fig. 5).

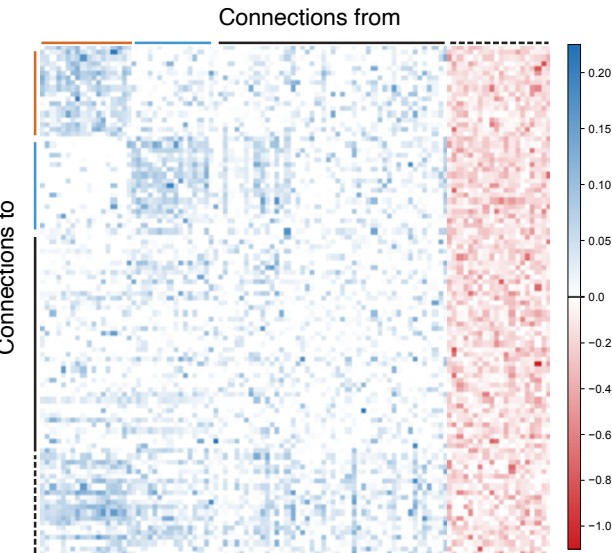

Figure 18: **Structure of $\mathbf{W}_{33}$ of $\mathbf{W}_{\text{rec}}$.** Full connectivity matrix of $\mathbf{W}_{33}$, reordered so that the structured excitatory components lie at the top left. The matrix is composed of a structured excitatory component (orange and blue), a set of random excitatory units (black), and a set of inhibitory units (dashed black), with non-obvious structure. The averaged connectivity matrix is shown in Fig. 6e.

# D Supplementary Notes

**Supplementary Note 1: Viewing the CB task as an XOR task**

Here we show that a nonlinearity is necessary to solve the task, proving that the task cannot be solved by the linear layer $\mathbf{W}_{\text{in}}$. First, we note that the Checkerboard task corresponds to an exclusive-or (XOR) problem. If we identify the two target configurations as 0 or 1 (corresponding to green on left, or green on right respectively, with the red target on the complement side), and the dominant checkerboard color as 0 or 1 (for green or red, respectively), then the output direction $d$ (identified as 0: left, 1: right) can be seen be in Table S1.

If the representation $\mathbf{r}$ was purely input driven, then:

$$\mathbf{r} = \mathbf{W}_{\text{in}}\mathbf{u}, \tag{11}$$

Our readout was a linear readout of the rates, i.e:

$$d = \mathbf{W}_{\text{out}}\mathbf{r} \tag{12}$$

The inputs $\mathbf{u}$ are the four dimensional input we trained with. But $\mathbf{u}$ is a linear transformation of two variables: the target orientation $\theta$, and checkerboard color $c$, which each can take two values. That is, if we let $\mathbf{q} = [\theta, c]$, then, the inputs could be written as a linear transformation of $\mathbf{q}$:

$$\mathbf{u} = \mathbf{W}\mathbf{q}, \tag{13}$$

where $\mathbf{W}$ is a linear transformation. Since the mappings from $\mathbf{q}$ to $d$ are all linear, they can be combined into a single linear transformation $\tilde{\mathbf{W}}$, i.e.,

$$d = \mathbf{W}_{\text{out}}\mathbf{W}_{\text{in}}\mathbf{W}\mathbf{q} = \tilde{\mathbf{W}}\mathbf{q}. \tag{14}$$

It is not possible for a linear classifier to solve the XOR problem by classifying correct outputs [51]. Hence, the trained RNNs cannot purely be input driven, and requires nonlinearity from the recurrent interactions to solve the task. After nonlinear processing, the left or right decision could be achieved by a linear readout of the units.

| target configuration | color | direction |
|:---:|:---:|:---:|
| 0 | 0 | 0 |
| 0 | 1 | 1 |
| 1 | 0 | 1 |
| 1 | 1 | 0 |

Table 2: Checkerboard task truth table

| context | signed color | signed motion | direction |
|:---:|:---:|:---:|:---:|
| 0 | 0 | 0 | 0 |
| 0 | 0 | 1 | 0 |
| 0 | 1 | 0 | 1 |
| 0 | 1 | 1 | 1 |
| 1 | 0 | 0 | 0 |
| 1 | 0 | 1 | 1 |
| 1 | 1 | 0 | 0 |
| 1 | 1 | 1 | 1 |

Table 3: Mante et al. [49] model truth table

## Supplementary Note 2: Mutual Information Estimation

The entropy of a distribution is defined as

$$H(x) = \mathbb{E}_{x \sim p(x)} \left[ \log \frac{1}{p(x)} \right]. \tag{15}$$

The mutual information, $I(X; Y)$, can be written in terms on an entropy term and as conditional entropy term:

$$I(Z; Y) = H(Y) - H(Y|Z). \tag{16}$$

We want to show that the usable information lower bounds the mutual information:

$$I(Z; Y) \geq I_u(Z; Y) := H(Y) - L_{CE}(p(y|z), q(y|z)) \tag{17}$$

It suffices to show that:

$$H(Y|Z) \leq L_{CE} \tag{18}$$

where $L_{CE}$ is the cross-entropy loss on the test set. For our study, $H(Y)$ represented the known distribution of output classes, which in our case were equiprobable.

$$H(Y|Z) := \mathbb{E}_{(z,y) \sim p(z,y)} \left[ \log \frac{1}{p(y|z)} \right] \tag{19}$$

$$= \underbrace{\mathbb{E}_{(z,y) \sim p(z,y)} \left[ \log \frac{1}{q(y|z)} \right]}_{\text{cross-entropy loss}} - \underbrace{\mathbb{E}_{z \sim p(z)} \left[ \text{KL}(p(y|z) || q(y|z)) \right]}_{\geq 0}, \tag{20}$$

$$\leq \mathbb{E}_{(z,y) \sim p(z,y)} \left[ \log \frac{1}{q(y|z)} \right] := L_{CE} \tag{21}$$

To approximate $H(Y|Z)$, we first trained a neural network with cross-entropy loss to predict the output, $Y$, given the hidden activations, $Z$, learning a distribution $q(y|z)$. The KL denotes the Kullback-Liebler divergence. We multiplied (and divided) by an arbitrary variational distribution, $q(y|z)$, in the logarithm of equation 19, leading to equation 20. The first term in equation 20 is the cross-entropy loss commonly used for training neural networks. The second term is a KL divergence, and is therefore non-negative. In our approximator, the distribution, $q(y|x)$, is parametrized by a neural network. When the distribution $q(y|z) = p(y|z)$, our variational approximation of $H(Y|Z)$, and hence approximation of $I(Z; Y)$ is exact [57–59].

In the paper, we additionally report the accuracy of the neural network on the test set. This differs from the cross-entropy in that the cross-entropy incorporates a weighted measure of the accuracy based on how "certain" the network is, while the accuracy does not.