# OpenReview forum: "A mechanistic multi-area recurrent network model of decision-making"
_NeurIPS.cc/2021/Conference — NeurIPS 2021 Poster_

### Official Review · Reviewer_vT5V · 2021-07-15

**Rating:** 4
**Confidence:** 4

**Summary:**

The authors train a RNN on a very simple neuroscience task that involves a binary decision based on the color content of two stimuli. The authors find that if they connect three RNNs in a similar feed-forward fashion as 3 specific brain areas involved in the task, the third RNN learns representations somewhat similar to what is found in a higher motor area (PMd). Specifically, their ‘area 3’ RNN discriminates the two choices (movement directions) but not color or context. This is only observed if 3 “areas” are trained and excitatory and inhibitory weights are separated, but not if only 1 or 2 areas were trained. They use the trained 3-area RNN to dissect the mechanisms behind the filtering of stimulus-related information while only the relevant choice information is passed on.


**Ethical Concerns:**

None.

**Limitations And Societal Impact:**

Yes.

**Main Review:**

This is a pure neuroscience analysis paper, with, as far as I could see, no real take home’s for machine learners. No novel methods were developed, not even formal details of the methods used were presented in the main text. Also, the analyses themselves have, as far as I could see, no implications for the design of AI/ML algorithms, and no attempt was made by the authors to work out any such potential implications.
Perhaps this is fine, but I guess it at least implies this paper can only be judged on grounds of whether it contributes something new to the body of knowledge in experimental neuroscience. I’m therefore not sure whether this conference is really the right fit for this type of paper, or whether a classical neuroscience journal would suit it better.

Putting this question aside, even within neuroscience, I’m not sure whether plain RNN training on behavioral tasks is, methodologically, still up to the state of the art (where the focus shifted to training RNNs directly on observed data in recent years). I’m also not sure how surprising the results really are: If color and context is directly fed into the first but not into the third RNN (cave: details were not clear from the main text to me), wouldn’t it be expected a priori that the third area discriminates less well between these properties than the first? And that if one has only one area for training, one finds that it unlike the 3-area RNN discriminates between all these features (as they were directly fed in)?
Also, to judge the biological plausibility of their 3-area vs. 1-area models the authors solely focused on this one property of whether color/context information can be read out or not in a given area. I would think that many more features of the neural activity across all three artificial areas would need to be compared to their biological counterparts to make a more convincing case for the model.

Finally, the authors stress a lot that, unlike in the biological preparation, they can dissect the RNN mechanisms in detail. But then a lot of the analyses rest on simple PCA-type projections that probably could have been done on experimental data as well. Admittedly, though, the analyses of the weights (Fig. 5,6) may have been more tricky, but perhaps correlation-based measures for deriving functional connectivity could have filled in.

The basic question is quite interesting, i.e. how neural systems learn to build parsimonious representations and to pass on to upstream areas only information needed to foster a decision. However, if that were the main question, I think it should have been addressed from a much wider angle, including more types of tasks and network designs, to allow for more generic conclusions and principles to be derived (beyond this one toy example). And a more rigorous mathematical and computational analysis would have been useful as well, given that – as the authors state themselves – a formal model of the process is available.

**Time Spent Reviewing:**

4

---

> ### Author Response · Authors · 2021-08-10
> **Response**
>
> We thank the reviewer for the comments and suggestions.
>
> > This is a pure neuroscience analysis paper, with, as far as I could see, no real take home’s for machine learners. No novel methods were developed, not even formal details of the methods used were presented in the main text. Also, the analyses themselves have, as far as I could see, no implications for the design of AI/ML algorithms, and no attempt was made by the authors to work out any such potential implications. Perhaps this is fine, but I guess it at least implies this paper can only be judged on grounds of whether it contributes something new to the body of knowledge in experimental neuroscience. I’m therefore not sure whether this conference is really the right fit for this type of paper, or whether a classical neuroscience journal would suit it better.
>
> Indeed, this is a computational neuroscience study using deep recurrent networks to better understand the principles underlying multi-area computation. In this manner, it provides an important framework that is currently lacking for experimental neuroscience to reason with recorded data from multiple areas. Concretely, we show how within-area dynamics can coordinate with inter-area connections to arrive at useful representations for doing the Checkerboard task. More broadly, and potentially of interest to deep learning, we found that relevant information was significantly preferentially aligned with the top singular vector of the feedforward matrices (and found that this was a consistent motif across nearly a hundred trained networks), and that incorporating biologically-inspired constraints can be important to learn minimal sufficient representations, a property often believed to be good representations (Alemi et al., 2016; Achille & Soatto, 2018). We also point out that Neurips invites Neuroscience and Cognitive Science papers in its call for papers (https://neurips.cc/Conferences/2021/CallForPapers).
>
> > Putting this question aside, even within neuroscience, I’m not sure whether plain RNN training on behavioral tasks is, methodologically, still up to the state of the art (where the focus shifted to training RNNs directly on observed data in recent years). I’m also not sure how surprising the results really are: If color and context is directly fed into the first but not into the third RNN (cave: details were not clear from the main text to me), wouldn’t it be expected a priori that the third area discriminates less well between these properties than the first? And that if one has only one area for training, one finds that it unlike the 3-area RNN discriminates between all these features (as they were directly fed in)? Also, to judge the biological plausibility of their 3-area vs. 1-area models the authors solely focused on this one property of whether color/context information can be read out or not in a given area. I would think that many more features of the neural activity across all three artificial areas would need to be compared to their biological counterparts to make a more convincing case for the model.
>
> The approach of training neural networks constrained by data versus training networks to perform a task are complementary approaches for using deep networks to understand neural data. The latter is a purely task-based constraint, and asks how does a standard neural network architecture, coupled with the constraints imposed by the task affect the dynamics and representations of the solution. The latter finds a normative solution to the task. In contrast, the former is a more constrained version, and requires experimental recordings from multiple areas, which are difficult to obtain. However, both methods have benefits, and the correct choice will depend on the particular setting. Our approach continues to be a widely used and important technique for training networks to gain insight into neural computation and mechanism discovery, and in this sense we believe it qualifies as “state of the art” (Yang et al., 2019; Banino et al., 2018; Orhan and Ma, 2019; and many other recent studies).
>
> Color and context were fed only into the first area. While it may be intuitively expected that color information is removed by Area 3, an alternative, and completely valid solution would be to retain all the semantically meaningful information (thus retaining color and direction information), and learn a decoder to readout the information. We showed this, as for example, 3-area networks trained without Dale’s law exhibited strong color representations in the last area (See Fig. 3a). It is therefore not the case that the color information is trivially absent in Area 3. Further, the high alignment of the direction-related activity with the top singular vector (Fig. 5) would be difficult to expect a priori.
>
> With respect to additional comparisons, we highlight that we did compare the distributions of choice probabilities for color and direction (the two behavioral outputs for this task) and found that Area 3 most closely resembled PMd data (Fig 2d, j). In addition to this, we performed CCA analysis on the principal components (Fig 2e, where we found Area 3 had the highest canonical correlation with PMd data). The PCs are commonly used as a summary of the neural population activity in a neural area, and relating the data PCs to RNN principal components has prior been a key metric for comparing RNNs to neurophysiology (Sussillo et al., 2015).  We also wish to emphasize that the primary phenomenon from the literature regarding this task was that color information was not filtered (Chandrasekaran et al., 2017, Wang et al., 2019), but importantly a mechanistic reason for why color information was not present has not been put forward. Many experimentalists have posed the mechanistic question of how such filtering is possible. Our modelling approach provides an explanation.
>
> > Finally, the authors stress a lot that, unlike in the biological preparation, they can dissect the RNN mechanisms in detail. But then a lot of the analyses rest on simple PCA-type projections that probably could have been done on experimental data as well. Admittedly, though, the analyses of the weights (Fig. 5,6) may have been more tricky, but perhaps correlation-based measures for deriving functional connectivity could have filled in.
>
> For Fig 5, 6, we need the weights to perform such analyses; and work attempting to infer connectivity from data is still nascent. Importantly, functional analyses also require difficult simultaneous recordings from multiple brain areas. Moreover, given the learned connectivity matrix, our results provide concrete predictions (alignment of relevant information w/ top singular vectors), and explain a previously only empirical result that certain task-related information was not present in the data (Chandrasekaran et al., 2017; Wang et al., 2019). We aren’t aware of another approach at the moment that could have arrived at the inter-area mechanism discovery we observed in a model consistent with PMd data.
>
> > The basic question is quite interesting, i.e. how neural systems learn to build parsimonious representations and to pass on to upstream areas only information needed to foster a decision. However, if that were the main question, I think it should have been addressed from a much wider angle, including more types of tasks and network designs, to allow for more generic conclusions and principles to be derived (beyond this one toy example). And a more rigorous mathematical and computational analysis would have been useful as well, given that – as the authors state themselves – a formal model of the process is available.
>
> RNNs, like the brain, are complicated nonlinear dynamical systems. Even though it is fully observable, it is oftentimes treated as a black box as they are often not easier to understand than biological neural networks. Here we show that by knowing the connections between the artificial neurons, we can expose relationships between within area dynamics and inter-area connections in task-optimized solutions with neuroscience-inspired constraints. We are interested and completely support work that provides more rigorous analyses of such systems, as well as extends these analyses to the more realistic multi-task setting. We emphasize that for these networks to be biologically relevant, they must bear resemblance to neurophysiological data and behavior. Towards this end, neuroscience behaviors and tasks are highly controlled, for reasons including the difficulty of training animals on more difficult tasks and controlling confounding factors that could explain computational hypotheses. Because our goal was to model tasks that are used in neuroscience (for which there is experimental data to ground the work), our RNNs are also trained on these tasks. We believe it is an important avenue for future work for experimentalists to move to more complex behavioral tasks and paradigms, and in turn for RNN models to also model these complex tasks and propose computational mechanisms for them.

---

> > ### Author Response · Authors · 2021-08-10
> > **Response (references)**
> >
> >
> > Achille, A., & Soatto, S. (2018). Emergence of invariance and disentanglement in deep representations. The Journal of Machine Learning Research, 19(1), 1947-1980.
> >
> > Alemi, A. A., Fischer, I., Dillon, J. V., & Murphy, K. (2016). Deep variational information bottleneck. arXiv preprint arXiv:1612.00410.
> >
> > Banino, A., Barry, C., Uria, B., Blundell, C., Lillicrap, T., Mirowski, P., ... & Kumaran, D. (2018). Vector-based navigation using grid-like representations in artificial agents. Nature, 557(7705), 429-433.
> >
> > Chandrasekaran, C., Peixoto, D., Newsome, W. T., & Shenoy, K. V. (2017). Laminar differences in decision-related neural activity in dorsal premotor cortex. Nature communications, 8(1), 1-16.
> >
> > Orhan, A. E., & Ma, W. J. (2019). A diverse range of factors affect the nature of neural representations underlying short-term memory. Nature neuroscience, 22(2), 275-283.
> >
> > Sussillo, D., Churchland, M. M., Kaufman, M. T., & Shenoy, K. V. (2015). A neural network that finds a naturalistic solution for the production of muscle activity. Nature neuroscience, 18(7), 1025-1033.
> >
> > Wang, M., Montanède, C., Chandrasekaran, C., Peixoto, D., Shenoy, K. V., & Kalaska, J. F. (2019). Macaque dorsal premotor cortex exhibits decision-related activity only when specific stimulus–response associations are known. Nature communications, 10(1), 1-16.
> >
> > Yang, G. R., Joglekar, M. R., Song, H. F., Newsome, W. T., & Wang, X. J. (2019). Task representations in neural networks trained to perform many cognitive tasks. Nature neuroscience, 22(2), 297-306.

---

> > > ### Comment · Reviewer_vT5V · 2021-08-31
> > > **thanks**
> > >
> > > I thank the authors for the clarification.
> > >
> > > I'm aware that Neurips also encourages contributions from cognitive and neuroscience, but even these - in my experience - usually combine this with novel methodological developments or insights, both of which are lacking from the present study.
> > > It may be debatable, but personally I also disagree with the statement that RNNs trained purely on a task are a complementary approach to RNNs trained on real data in  addition. In computational neuroscience in my understanding you want to obtain insights into how the real brain works, and for that training on actual recordings and behavior I think is the much stronger approach. Since it makes a much tighter link to actual empirical data, it renders more confidence into the biological significance of the mechanisms and properties found through model training.
> > > Finally, if trained RNNs are indeed not easier to understand than the biological substrate (as the authors indicate in their last response), I wouldn't see what's the point of using them for tackling brain mechanisms.
> > >
> > > I think though the additional experiments performed make at least a somewhat more convincing case for the importance of 3 areas and Dale's law, and so are certainly a step into the right direction. If these conclusions could be supported even further, and also the link to actual neural recordings could be extended, I think this could be a great computational study. I certainly agree here with the other referees that looking at information propagation and filtering across consecutive brain areas is an interesting and very worthwhile question from a neuroscience angle.

---

### Official Review · Reviewer_EDPH · 2021-07-16

**Rating:** 4
**Confidence:** 4

**Summary:**

This study examines a "checkerboard" task, in which subjects had to determine the dominant color of a stimulus to select one out of two possible targets, the position of which (="context") varied across trials. Neurophysiological recordings in the premotor cortex of monkeys trained on this task revealed that neural activity encoded only the final decision, but not the sensory information on the color of the stimulus, or position of the targets. To investigate potential mechanisms, this study examines multi-area recurrent neural networks trained on the task. The activity in the different areas is analysed using population methods from systems neuroscience, in particular targeted dimensionality reduction and decoding.
The main result is that the network reproduces the main experimental observation, the fact that only decision is represented in the last area. Moreover, training experiments indicate that at least three areas, and segregation between excitation and inhibition, are needed to reproduce this observation.

**Limitations And Societal Impact:**

yes

**Main Review:**

This study is a nice application of a variety of methods from neuroscience  to analyse trained RNNs. The main result, the fact that only output-relevant information is present in the last area, is interesting. While the analysis is extensive, it is rather descriptive, and ultimately the underlying mechanism unfortunately remains unclear.

In particular, the main finding that at least three areas are needed is empirical, but it is unclear why two areas would not be enough to eliminate all information other than output. Indeed the analyses of the first area (Fig 4) show that direction and color are already separated along orthogonal axes at that stage. So projecting onto the direction axis would be sufficient to eliminate all other information. One additional area would be sufficient to do this projection. Why are three areas needed ?

Similarly, it is unclear why the segregation between excitation and inhibition would be needed, it seems that a network without Dale's law should be able to perform the same projection. (Detail: the text mentions the importance of feed-forward inhibition, but it is not clear what is meant by that, inhibition across areas?)

Finally, the experimentally testable hypotheses announced in the abstract do not seem to be fully developed.

Altogether, this study falls somewhat short of the promised mechanistic understanding.

**Time Spent Reviewing:**

4 hours

---

> ### Author Response · Authors · 2021-08-10
> **Response**
>
> We thank the reviewer for the comments and suggestions.
>
> > This study is a nice application of a variety of methods from neuroscience to analyse trained RNNs. The main result, the fact that only output-relevant information is present in the last area, is interesting. While the analysis is extensive, it is rather descriptive, and ultimately the underlying mechanism unfortunately remains unclear.
>
> We appreciate the reviewer's perspective that the underlying mechanism for how only output-relevant information is present in the last area is unclear. We do admit that the choice of using the word mechanistic is subjective and potentially contentious. However, here -- by mechanistic -- we are referring to how the artificial unit activity is related to their connections, a feature that is not typically available in experimental neural data. We will clarify this in the revision of our manuscript. Through this approach, we make several important contributions. First, we find that the output-relevant (direction) information is preferentially aligned with the top singular values of the feedforward matrix, whereas the color information was randomly aligned. This result was robust (across nearly a hundred networks). An alternative, and potentially equally possible solution, would be for the irrelevant information to be preferentially filtered (and be aligned to the lowest singular values) but this is not what we observed. We hope our responses will further clarify this.
>
> > In particular, the main finding that at least three areas are needed is empirical, but it is unclear why two areas would not be enough to eliminate all information other than output. Indeed the analyses of the first area (Fig 4) show that direction and color are already separated along orthogonal axes at that stage. So projecting onto the direction axis would be sufficient to eliminate all other information. One additional area would be sufficient to do this projection. Why are three areas needed ?
>
> The primary mechanism through which only direction information is present by the last area was through the preferential alignment of the direction axis with the top singular vectors of the feedforward matrix. Importantly, our network did not actively suppress the irrelevant information, it merely treated it almost equivalently to a random axis of variability (Fig 5). That is the color axis did not preferentially align with the lowest singular values, but was aligned similar to a random axis. In comparison to 2-area RNNs, 3-area RNNs have additional processing stages that can 1) further attenuate irrelevant information, and 2) amplify relevant information. Moreover, a 2-area RNN with input color information should require a more complicated readout from the second area in order to demix the direction information and color information. We therefore ran three analyses: (1) we assessed if color information (and therefore input information) was propagated more in 2-area vs 3-area RNNs, (2) we assessed the readout complexity of the output in 2-area vs 3-area RNNs, and (3) we assessed if Area 2 of the 3-area RNN played a role in amplifying relevant information.
>
> For (1), we re-ran the feedforward analyses on the two area network (on eight different random seeds), and  found that the direction axis was not as preferentially aligned with the top singular vectors as in the three area network (with an average potent projection of ~0.15 onto the top singular vector as opposed to 0.6 for the 3-area networks). The color axis was more preferentially propagated than in the three area-networks. (We will include this new figure in the appendix.) This suggests that the 2-area network arrived at a solution that propagated the input information and used a more complicated readout (since performance was equivalent). For (2), we assessed the readout of the last area in 2-area and 3-area RNNs. We found that while 3-area RNNs had separate pools to readout right vs left decisions (Fig 6b), 2-area RNN units had weights for both the left and right decisions, increasing the readout complexity. For (3), we analyzed Area 2 of the 3-area RNN and found that the area played a strong role in amplifying differences in direction representations relative to color representations, but the 2-area RNN did not have such an area. We will incorporate these points into the revised manuscript.
>
> In summary, for a 2-area RNN to solve the task in the same manner as the 3-area RNN, we believe the training will need to find a solution that has the following properties: 1) Alignment of the direction-axis with the strongest singular vector(s) of the connectivity matrix, 2) Alignment of the color-axis as much as possible to the weakest singular vector(s) of the connectivity matrix. The advantage that the 3- and 4-area RNNs have is that they can use the intermediate areas (area 2, and area 2, 3 for 3 and 4-area RNNs respectively) for amplifying direction information and suppressing color information by aligning color-related information with random directions and thus filtering color information for subsequent areas, as opposed to needing to learn to align such information with the lowest singular vectors.
>
> > Similarly, it is unclear why the segregation between excitation and inhibition would be needed, it seems that a network without Dale's law should be able to perform the same projection. (Detail: the text mentions the importance of feed-forward inhibition, but it is not clear what is meant by that, inhibition across areas?)
>
> Indeed, an alternative solution would be for the standard unconstrained three area network to learn a representation that only contained direction information. Prompted by the comment, we performed the same analyses of the three area unconstrained networks (similar to Fig. 5) and found that direction information direction information was only slightly more propagated than color information, while color was propagated at near chance levels (We will add this figure to the Appendix). This suggests that the networks learned a solution that retained all input information, and learned an appropriate readout decoder. Dale’s law imposed a constraint, and this constraint was sufficient to lead to a solution that only retained the relevant information. Of note, in deep learning optimal representations are thought to be minimal and sufficient (Achille & Soatto, 2018), and that standard deep network training (which includes explicit and implicit regularization from training with SGD) lead to such solutions. Thus it is possible that there are some machine learning constraints, which when applied to a  multi-area RNN, would lead to similar solutions as the networks with Dale’s law. Thus Dale’s law (and the constraint imposed) should be interpreted as a sufficient constraint: one that enables an understanding for how the sensory information could be transformed to a direction decision (using a representation that does not include the sensory information, like in PMd). We will add this to the discussion of the manuscript.
>
> The feedforward inhibition refers to inhibition across areas, e.g., between Area 1 and 2, it refers to the excitatory connections from excitatory neurons in Area 1 to inhibitory neurons in Area 2. We will be sure to make this clear in the main text.
>
> > Finally, the experimentally testable hypotheses announced in the abstract do not seem to be fully developed.
>
> "Our analysis of the multi-area RNN leads to testable hypotheses for future experiments. First, we expect that neurons in cortical areas upstream of PMd should exhibit mixed selectivity for color and direction information, consistent with studies of dorsolateral prefrontal cortex (DLPFC) and ventrolateral prefrontal cortex (VLPFC) in cognitive tasks (Rigotti et al, 2013, Mante et al., 2013, Fusi et al., 2016). More specifically, our model predicts the following organization of population dynamics in these areas: neural population dynamics should diverge to two regions with slow dynamics based on target configuration, with largely overlapping context and color axes, but an orthogonal direction axis. Second, due to alignment of inter-area connections, direction axis activity in DLPFC/VLPFC should be more predictive of activity in downstream regions such as PMdr and PMd than activity in the top PCs. We anticipate that communication subspace analyses identified in recent papers should help test these predictions (Semedo et al., 2019; Kohn et al., 2020).
>
> Additionally, in the output area of our RNN, our connectivity matrices indicate that inhibitory connections play an important role in mutually inhibiting clusters of neurons that accumulate evidence for a direction decision. When we selectively increased the inhibition to a cluster, we observed that the opposite cluster was more likely to win the competition. We hypothesize that injecting muscimol in areas of PMd should selectively bias the monkey towards one or the other direction of reaching. Similarly, in tasks that involve bilateral brain networks, muscimol in one hemisphere should bias behavior towards the contralateral side.”
> - - -
> Achille, A., & Soatto, S. (2018). Emergence of invariance and disentanglement in deep representations. The Journal of Machine Learning Research.
>
> Mante, V., Sussillo, D., Shenoy, K. V., & Newsome, W. T. (2013). Context-dependent computation by recurrent dynamics in prefrontal cortex. nature.
>
> Kohn, A., Jasper, A. I., Semedo, J. D., Gokcen, E., Machens, C. K., & Byron, M. Y. (2020). Principles of corticocortical communication: proposed schemes and design considerations. Trends in Neurosciences.
>
> Rigotti, M., Barak, O., Warden, M. R., Wang, X. J., Daw, N. D., Miller, E. K., & Fusi, S. (2013). The importance of mixed selectivity in complex cognitive tasks. Nature, 497(7451), 585-590.
>
> Semedo, J. D., Zandvakili, A., Machens, C. K., Byron, M. Y., & Kohn, A. (2019). Cortical areas interact through a communication subspace. Neuron, 102(1), 249-259.

---

> > ### Comment · Reviewer_EDPH · 2021-08-31
> > **thank you**
> >
> > Thank you very much for a detailed reply.

---

### Official Review · Reviewer_L9GC · 2021-07-16

**Rating:** 8
**Confidence:** 4

**Summary:**

The authors train a multi-area (modular) RNN to do a decision-making task known as the "Checkerboard" task and compare the RNN to recorded neurons in dorsal premotor cortex (PMd). They 1) show how the final layer of the multi-area RNN better match PMd responses compared with a standard RNN; 2) Investigate different parameter/architecture regimes of the multi-area RNN to determine what type of network is important for producing PMd-like responses; and 3) investigate the mechanisms by which information is (or isn't) transmitted through the multi-area RNN.

**Limitations And Societal Impact:**

Limitations are briefly mentioned, but could be mentioned more explicitly or expanded on.

Potential negative societal impacts are not mentioned in the text

**Main Review:**

Pros:

-The manuscript is very clearly written.

-The manuscript is extremely thorough and high quality - the authors provide a plethora of analyses in the main text and in the appendix. There is definitely enough here to be a full journal article.

-Using RNNs, the paper investigates a significant question in neuroscience - how multiple regions may work together to produce behavior.


Cons:

-The paper is not very novel in a machine learning sense and multi-area RNNs have been used within neuroscience (although only recently). However, I'm not overly concerned about this given that the paper uses an existing tool (multi-area RNNs) to do many interesting analyses.


Minor comments:

Line 128: "is contained is" -> "is contained in"

What happens if you train the RNN without Dale's law? Do you still get PMd-like representations (as in Fig. 3A), but you wanted to use Dale's law to understand what parameters work for this type of network?

Line 184: Fig 4e isn't a panel.

You say "see Methods" a couple of times, but this should be referring to the Appendix.

Fig. 5 legend: You say the "color axis is more aligned with the nullspace." From this figure, it's clear to me that the color axis is not aligned with the potent space, but having random alignment doesn't necessarily seem like nullspace alignment to me (i.e. information is not actively being filtered out). This may all just be a matter of semantics, but I wanted to point it out.

Line 195: Are your findings still the same for all connections (not just limited to E->E)? Either finding would be interesting.

Does that Area 3 mechanism just boost decoding accuracy from ~95 to ~100% ?
-Seems like already linearly separable, so what's the point of this mechanism?

**Time Spent Reviewing:**

7

---

> ### Author Response · Authors · 2021-08-10
> **Response**
>
> We thank the reviewer for the comments and suggestions. We will address the minor comments below.
>
> > What happens if you train the RNN without Dale's law? Do you still get PMd-like representations (as in Fig. 3A), but you wanted to use Dale's law to understand what parameters work for this type of network?
>
> We don’t observe PMd-like representations (see Fig 3a, where we swept a 3 area recurrent networks without Dale’s law at various feedforward sparsity constraints), since color information is present. In this manner, our modeling study is one where imposing a biological constraint led to a more biologically plausible model. It also enabled us to unpack mechanisms behind such biologically-plausible solutions.
>
> > Fig. 5 legend: You say the "color axis is more aligned with the nullspace." From this figure, it's clear to me that the color axis is not aligned with the potent space, but having random alignment doesn't necessarily seem like nullspace alignment to me (i.e. information is not actively being filtered out). This may all just be a matter of semantics, but I wanted to point it out.
>
> This is a subtle but important point, which we had previously clarified in the main text. We will also clarify the legend -- thank you for pointing it out.
>
> > Line 195: Are your findings still the same for all connections (not just limited to E->E)? Either finding would be interesting.
>
> This is actually a key parameter that affects our results (See Fig 3b). A limited E-I connectivity seems to impose a strong constraint so that the network settles at a solution with only output-relevant information. With increasing E-I, there was increasing color information (Fig 3b). Note that our parameter choices for the inter-area E-I connectivity are consistent with anatomical evidence, which suggests approximately 10% of feedforward connections are E-I (Barbas, 2015)
>
> > Does that Area 3 mechanism just boost decoding accuracy from ~95 to ~100% ? -Seems like already linearly separable, so what's the point of this mechanism?
>
> The winner-take all mechanism in Area 3 is a natural solution for a network to produce evidence for one reach and little evidence for the other reach (Wong and Wang, 2016). It enables a simple readout of a decision from a pool of neurons, as opposed to linearly weighting all neurons. It’s difficult, however, to precisely quantify the change in accuracy that this mechanism provides.
>
> - - -
> Barbas, H. (2015). General cortical and special prefrontal connections: principles from structure to function. Annual review of neuroscience, 38, 269-289.
>
> Wong, K. F., & Wang, X. J. (2006). A recurrent network mechanism of time integration in perceptual decisions. Journal of Neuroscience, 26(4), 1314-1328.

---

> > ### Comment · Reviewer_L9GC · 2021-08-30
> > **Thank you for your responses**
> >
> > Thank you for your detailed responses to myself and all the other reviewers.  Just so you know, given the differences of opinion, we (all the reviewers) have been discussing over the last couple weeks - sorry that you have been waiting in the dark.
> >
> > I agree with the other reviewers' comments that this paper could be made stronger by understanding the mechanism by which including Dale's law leads to PMd-like responses, and also by slightly reducing the claims (e.g. the abstract's statement that "incorporating multiple areas and Dale's Law is critical"  -  given that Dale's law is likely not the only way to get PMd-like responses).

---

### Official Review · Reviewer_jJXH · 2021-07-17

**Rating:** 7
**Confidence:** 4

**Summary:**

This paper uses RNNs to study the role of PMd in a decision making task. The paper finds that representations in PMd are informative about the output (direction of movement), but not about the inputs to the task (here, the color information). As this information is necessary to solve the task, it suggests that areas prior to PMd represent this information. The paper builds of a model of this multi-area computation using stacked recurrent networks. For certain settings of architecture parameters, the final area in the network resembles PMd (in that the final area contains direction, but not color, information). The paper probes this multi-area RNN to understand how it solves the task.

**Main Review:**

I thought this was a really clever use of multi-area (stacked) RNNs to shed insight onto a complex task (decision making) that involves multiple brain areas, as well as a really effective use of dimensionality reduction (particularly, dPCA) to reverse engineer a trained network.

Some minor concerns/questions:
- What about the dynamics of Area 2? What computation(s) are they responsible for? If they are not necessary, then how come you need 3 areas before the last one resembles PMd?
- Another lens on to the dynamics of RNNs is to linearize them around fixed points of the dynamics, as in [Sussillo and Barak 2013](https://direct.mit.edu/neco/article/25/3/626/7854/Opening-the-Black-Box-Low-Dimensional-Dynamics-in). I would be curious to see if using that kind of analysis on each of the areas in your network led to any additional insight. For example, you might be able to better identify or quantify the winner-take-all dynamics in Area 3 by looking for approximate fixed points of the dynamics of the area 3 RNN.
- If one really believes that the multi-area RNN corresponds to multiple brain areas, then this work provides strong predictions for what to expect if we were to record from these additional areas during the same task. These predictions could be more explicitly spelled out in the discussion (even if you are agnostic about what brain area(s) the RNN Areas 1 & 2 correspond to).
- Can this multi-area RNN approach help inform future PMd experiments? For example by helping shape what kinds of tasks we should train monkeys on in order to find or highlight interesting computations in PMd?

**Time Spent Reviewing:**

2.5

---

> ### Author Response · Authors · 2021-08-10
> **Response**
>
> We thank the reviewer for the comments and suggestions. We will address the minor comments below.
>
> > What about the dynamics of Area 2? What computation(s) are they responsible for? If they are not necessary, then how come you need 3 areas before the last one resembles PMd?
>
> Area 2 acts as an intermediary, serving as a relative amplifier where it amplifies the direction information (difference in activity between left and right choices) relative to color information (difference in activity between red and green choices). It does this through both within area dynamics in Area 2 and the inter-area connections between the second and third area. Moreover, when we analyzed the 2 area networks similar to Fig 5 (across eight random seeds), we found that the direction axis was not as preferentially aligned with the feedforward connectivity matrix between Area 1 and 2 (with an average potent projection of ~0.15 onto the top singular vector as opposed to 0.6 for the 3-area networks). This suggests that the network settled in a different solution; one where all input information was propagated to the output area, as opposed to only the direction information. We will include these updated analyses in the Appendix.
>
> > Another lens on to the dynamics of RNNs is to linearize them around fixed points of the dynamics, as in Sussillo and Barak 2013. I would be curious to see if using that kind of analysis on each of the areas in your network led to any additional insight. For example, you might be able to better identify or quantify the winner-take-all dynamics in Area 3 by looking for approximate fixed points of the dynamics of the area 3 RNN.
>
> We agree that fixed point analyses provide a complementary lens to understand dynamics in recurrent networks, however, analyzing the fixed points for multi-area networks requires care, since standard fixed point analyses are done with constant inputs. In multiple area networks, choosing the constant input is not obvious, since they come from other areas (for example the input from Area 2 to 3 is not constant). We agree that this is a complementary approach that can lead to additional insights, especially for understanding the within-area dynamics, and is interesting future work.
>
> > We have included the following direct testable hypotheses, which we will include in the main text:
>
> "Our analysis of the multi-area RNN leads to testable hypotheses for future experiments. First, we expect that neurons in cortical areas upstream of PMd should exhibit mixed selectivity for color and direction information, consistent with studies of dorsolateral prefrontal cortex (DLPFC) and Ventrolateral Prefrontal cCortex (VLPFC) in cognitive tasks (Rigotti et al, 2013, Mante et al., 2013, Fusi et al., 2016). More specifically, our model predicts the following organization of population dynamics in these areas: nNeural population dynamics should diverge to two regions with slow dynamics based on target configuration, with largely overlapping context and color axes, but an orthogonal direction axis. Second, due to alignment of inter-area connections, direction axis activity in DLPFC should be more predictive of activity in downstream regions such as PMdr and PMd than activity in the top PCs." We anticipate that communication subspace analyses identified in recent papers should help test these predictions (Semedo et al., 2019; Kohn et al., 2020).
>
> Additionally, in the output area of our RNN, our connectivity matrices indicate that inhibitory connections play an important role in mutually inhibiting clusters of neurons that accumulate evidence for a direction decision. When we selectively increased the inhibition to a cluster, we observed that the opposite cluster was more likely to win the competition. We hypothesize that injecting muscimol in areas of PMd should selectively bias the monkey towards one or the other direction of reaching. Similarly, in tasks that involve bilateral brain networks, muscimol in one hemisphere should bias behavior towards the contralateral side.”
>
> > Can this multi-area RNN approach help inform future PMd experiments? For example by helping shape what kinds of tasks we should train monkeys on in order to find or highlight interesting computations in PMd?
>
> Our modelling approach, coupled with the experimental data from PMd, suggest that only motor-related (also referred to as action) signals are present in later brain areas, such as PMd. To better understand the role of PMd during perceptual decision-making tasks, an interesting experiment would be one that biases the reaches towards one target by making those reaches easier. In this manner, the movements become more diverse, and the PMd representations may change depending on cost/difficulty of performing a particular reach.
>
> - - -
> Mante, V., Sussillo, D., Shenoy, K. V., & Newsome, W. T. (2013). Context-dependent computation by recurrent dynamics in prefrontal cortex. nature, 503(7474), 78-84.
>
> Fusi, S., Miller, E. K., & Rigotti, M. (2016). Why neurons mix: high dimensionality for higher cognition. Current opinion in neurobiology, 37, 66-74.
>
> Kohn, A., Jasper, A. I., Semedo, J. D., Gokcen, E., Machens, C. K., & Byron, M. Y. (2020). Principles of corticocortical communication: proposed schemes and design considerations. Trends in Neurosciences.
>
> Rigotti, M., Barak, O., Warden, M. R., Wang, X. J., Daw, N. D., Miller, E. K., & Fusi, S. (2013). The importance of mixed selectivity in complex cognitive tasks. Nature, 497(7451), 585-590.
>
> Semedo, J. D., Zandvakili, A., Machens, C. K., Byron, M. Y., & Kohn, A. (2019). Cortical areas interact through a communication subspace. Neuron, 102(1), 249-259.

---

> > ### Comment · Reviewer_jJXH · 2021-08-30
> > **Thank you for your response**
> >
> > Thank you for your response. I would love to see further analysis of the RNN (especially RNNs with different numbers of layers, and different networks that do or do not observe Dale's law) to better understand why these architectural features are necessary to match the experimental data,

---

> > > ### Author Response · Authors · 2021-09-02
> > > **Thanks for the reply**
> > >
> > > Thanks for the reply. We agree and had performed additional analyses, also prompted by Reviewer EDPH.
> > >
> > > **In response to the number of areas:** The primary mechanism through which only direction information is present by the last area was through the preferential alignment of the direction axis with the top singular vectors of the feedforward matrix. Importantly, our network did not actively suppress the irrelevant information, it merely treated it almost equivalently to a random axis of variability (Fig 5). That is the color axis did not preferentially align with the lowest singular values, but was aligned similar to a random axis. In comparison to 2-area RNNs, 3-area RNNs have additional processing stages that can 1) further attenuate irrelevant information, and 2) amplify relevant information. Moreover, a 2-area RNN with input color information should require a more complicated readout from the second area in order to demix the direction information and color information. We therefore ran three analyses: (1) we assessed if color information (and therefore input information) was propagated more in 2-area vs 3-area RNNs, (2) we assessed the readout complexity of the output in 2-area vs 3-area RNNs, and (3) we assessed if Area 2 of the 3-area RNN played a role in amplifying relevant information.
> > >
> > > For (1), we re-ran the feedforward analyses on the two area network (on eight different random seeds), and found that the direction axis was not as preferentially aligned with the top singular vectors as in the three area network (with an average potent projection of ~0.15 onto the top singular vector as opposed to 0.6 for the 3-area networks). The color axis was more preferentially propagated than in the three area-networks. (We will include this new figure in the appendix.) This suggests that the 2-area network arrived at a solution that propagated the input information and used a more complicated readout (since performance was equivalent). For (2), we assessed the readout of the last area in 2-area and 3-area RNNs. We found that while 3-area RNNs had separate pools to readout right vs left decisions (Fig 6b), 2-area RNN units had weights for both the left and right decisions, increasing the readout complexity. For (3), we analyzed Area 2 of the 3-area RNN and found that the area played a strong role in amplifying differences in direction representations relative to color representations, but the 2-area RNN did not have such an area. We will incorporate these points into the revised manuscript.
> > >
> > > In summary, for a 2-area RNN to solve the task in the same manner as the 3-area RNN, we believe the training will need to find a solution that has the following properties: 1) Alignment of the direction-axis with the strongest singular vector(s) of the connectivity matrix, 2) Alignment of the color-axis as much as possible to the weakest singular vector(s) of the connectivity matrix. The advantage that the 3- and 4-area RNNs have is that they can use the intermediate areas (area 2, and area 2, 3 for 3 and 4-area RNNs respectively) for amplifying direction information and suppressing color information by aligning color-related information with random directions and thus filtering color information for subsequent areas, as opposed to needing to learn to align such information with the lowest singular vectors.
> > >
> > > **In response to the effect of Dale's law, and whether a network without Dale's law could arrive at a similar solution:** Indeed, an alternative solution would be for the standard unconstrained three area network to learn a representation that only contained direction information. Prompted by the comment, we performed the same analyses of the three area unconstrained networks (similar to Fig. 5) and found that direction information direction information was only slightly more propagated than color information, while color was propagated at near chance levels (We will add this figure to the Appendix). This suggests that the networks learned a solution that retained all input information, and learned an appropriate readout decoder. Dale’s law imposed a constraint, and this constraint was sufficient to lead to a solution that only retained the relevant information. Of note, in deep learning optimal representations are thought to be minimal and sufficient (Achille & Soatto, 2018), and that standard deep network training (which includes explicit and implicit regularization from training with SGD) lead to such solutions. Thus it is possible that there are some machine learning constraints, which when applied to a multi-area RNN, would lead to similar solutions as the networks with Dale’s law. Thus Dale’s law (and the constraint imposed) should be interpreted as a sufficient constraint: one that enables an understanding for how the sensory information could be transformed to a direction decision (using a representation that does not include the sensory information, like in PMd). We will add this to the discussion of the manuscript.
> > >
> > > We hope these additional analyses better clarify our contributions.

---

### Decision · Program_Chairs · 2021-09-28

**Decision:**

Accept (Poster)

**Comment:**

The reviewers appreciated the clear presentation of the paper, the interesting observation that a three-area network can qualitatively reproduce the results of the Mante et al paper with regards to neural selectivity, and the thorough investigation of different hyper-parameters influencing network behaviour. However, and after extensive internal discussions in which all reviewers participated (even if they might not have updated their review reports), it remained unclear what the _causes_ of this observation were-- in part, the paper mostly (and in contrast to some of its claims) mostly made statements about _sufficient_ ingredients of the networks, and not about _necessary_ ones (e.g. regarding the role of Dale's law). More broadly, the study would have benefited from a clearer statement about _why_ a  three-area (and not, e.g. a two-area) network can achieve the desired behavior.  We hope that the feedback by the reviewers will allow be useful for you.

**Consistency Experiment:**

NeurIPS has a long history of experimentation. In 2014, NeurIPS ran an experiment in which 10% of submissions were reviewed by two independent committees to quantify the randomness in the review process. This year, we repeated a variant of this experiment to see how the quality of the review process has changed over time.  This paper was part of the experiment and was therefore assigned to two committees (consisting of reviewers, an Area Chair, and a Senior Area Chair) that reached independent decisions.  If both committees made the same recommendation, this recommendation was followed. If a single committee recommended acceptance, the paper was accepted (with the exception of a few cases in which the other committee identified what we considered a fatal flaw, e.g., an error in a key result).

This copy’s committee reached the following decision: **Reject**

The other committee assigned to the paper recommended **Accept (Poster)**.  You can find the other set of reviews, along with any follow up discussion with the authors here:
https://openreview.net/forum?id=NENYf2nxnrR